# Blending Neural Control Density Functions for Stabilization and Safety

**Sahil Chaudhary** [* 1]   **Chaitanya Murti** [* 2]   **Chiranjib Bhattacharyya** [1]

## Abstract

Recent work on Neural Network-based methods for nonlinear control use Lyapunov Functions to obtain controllers with guarantees of stability. However, Lyapunov-based methods are fundamentally limited: they cannot be used for smooth blending with formal Region of Attraction (RoA) expansion guarantees, and also fail to certify stability when unstable equilibria or saddle points are present. Density functions provide an alternate stability certificate, and address these limitations by certifying almost everywhere stability, and enable smooth blending of controllers. Learning valid density certificates is challenging due to integrability constraints, and the effect of density-based blending controllers on RoAs is not well understood. In this work, we provide the first guarantee that controllers blended with density functions yield RoAs containing the union of the RoAs achieved by the constituent controllers. Then, we propose a novel exponential characterization of density functions that provably satisfies the integrability condition, and introduce Neural Control Density Functions (NCDFs), that leverage this new parameterization. We also extend NCDFs for synthesizing safe-stable controllers by combining NCDFs with control barrier functions (NCDF-CBFs). Our experiments[1] show that blended controllers obtain superior RoAs to state-of-the-art methods like Neural Lyapunov Control and Sum-of-Squares based techniques.

## 1. Introduction

Neural network-based methods for nonlinear control in robotics have attracted significant interest in recent years, with applications in underactuated robotics (Chang et al., 2019; Yang et al., 2024; Wu et al., 2023). These work enable searching for stabilizing controllers, and Lyapunov functions to certify stability simultaneously. These methods, which are instances of counterexample guided inductive synthesis (CEGIS) methods (Abate et al., 2018), combine efficient gradient-based optimization techniques for control synthesis along with a formal verifier, rather than the expensive SDP- or LP-based techniques traditionally used in methods such as Sum-of-Squares techniques (Parrilo, 2000).

However, simply providing certificates of stability is often insufficient, with many real-world applications requiring additional guarantees, like safety or large regions of attraction (Dawson et al., 2023; Kant et al., 2019). Using Lyapunov functions for stability analysis does have drawbacks; for instance, as shown in Prieur & Praly (1999), different controller/Lyapunov function pairs cannot be combined to produce a more powerful controller. Similarly, Lyapunov functions cannot certify stability for systems with multiple stable and unstable equilibria (Khalil; Rantzer, 2001).

Density functions, first introduced in Rantzer (2001), offer solutions to many of these issues. Since they certify *almost everywhere* (a.e.) stability that is, the density function certifies that almost all trajectories converge to the equilibrium, it has been shown that they can certify a.e. stability even when the system has unstable equilibria (Rantzer, 2001). The density function stability criterion also possesses a convexity property, allowing for joint synthesis of the controller and the density function (Rantzer, 2001; Prajna, 2003), which could not be achieved with Control Lyapunov functions (CLFs), as noted in Prieur & Praly (1999). Density functions also enable combining, or *blending*, of controllers and certificates (Rantzer & Ceragioli, 2001), which otherwise has only been possible when controllers either share a common CLF or when a controller has multiple admissible CLFs (Grammatico et al., 2013; Bianchini et al., 2018). However, the extent to which blending controllers and stability certificates increases the region of attraction remains an open question,

---

[*]Equal contribution  [1]Indian Institute of Science, Bengaluru, India  [2]HP Inc. AI Lab, Bengaluru, India. Correspondence to: Sahil Chaudhary <sahilc@iisc.ac.in>, Chaitanya Murti <chaitanya.murti@hp.com>, Chiranjib Bhattacharyya <chiru@iisc.ac.in>.

*Proceedings of the $43^{rd}$ International Conference on Machine Learning*, Seoul, South Korea. PMLR 306, 2026. Copyright 2026 by the author(s).

[1]Our code is available at **this GitHub repository** .

and synthesizing controllers using density functions has typically required solving expensive sum-of-squares programs (Prajna et al., 2004).

In this work, we address three key questions. First, can we use neural networks in the CEGIS framework to certify the a.e. stability of nonlinear systems, while accounting for unstable equilibria? Second, to what extent do blended controllers outperform the constituent controllers? Last, can density functions be used to synthesize certifiably safe and stable controllers using neural networks? We state our contributions formally in the sequel. To that end, we propose a novel exponential parametrization of the density functions satisfying a crucial integrability condition and that can be implemented with neural networks. We also show that these controllers can be blended, and provide the first formal analysis of the region of attraction of a blended controller. Last, we provide a method for safe and stable control synthesis using neural density functions.

**Analysis and Control with Learning Neural Control Density Functions**   Motivated by the fact that Lyapunov functions often fail to certify stability in the presence of multiple equilibria, we propose **Neural Density Functions** (NDFs). In order to learn valid NDFs, we propose a novel exponential parameterization that uses two neural networks to construct the density function, that guarantees the key integrability requirement stated in (Rantzer, 2001), which we prove in Propostion 5.1. We then develop the Neural Control Density Functions (NCDFs), and propose a CEGIS-based algorithm for learning NDFs and NCDFs Algorithm 2. We demonstrate the efficacy of our method in control synthesis, and stability instability analysis on several benchmarks, notably showing the improvement of Neural Densify Functions over Lyapunov-based approaches in systems with multiple equilibria.

**Blending Controllers with Density Functions**   Motivated by the challenge of obtaining controllers with large regions of attraction, we leverage density functions to *blend* controllers, resulting in closed-loop systems with superior regions of attraction than those of the constituent dynamics. We first provide a novel characterization of the almost-everywhere regions of attraction that can be certified by density functions in Theorem 4.1. The key challenge is then to rigorously quantify the improvement in the region of attraction attained by blending controllers. In Theorem 4.4, we provide the first result showing that the regions of attraction of blended controllers contain the union of the individual regions of attraction. We show empirically that the regions of attraction obtained by blending controllers indeed contain the union of the RoAs of the constituent controllers, thereby guaranteeing improvement over all baselines.

**Safe Stabilization with NCDFs**   Synthesizing safe and stabilizing controllers is an active area of research. In Theorem 6.1, we combine the stability conditions in (Rantzer, 2001) with control barrier functions stated in (Ames et al., 2019) to characterize controllers that are safe and stable with density functions, and that the blending property applies to safety as well. We show how this result can be used to synthesize NCDF-based controllers that are both safe and stable, and demonstrate their efficacy on several benchmarks.

## 2. Related Work

In this section, we discuss some relevant related work and how our work contrasts with them.

**Neural Networks in Control**   Using neural networks for control with stability guarantees was first proposed in (Chang et al., 2019), which trained a neural network on trajectory data and used an SMT solver to generate counterexamples; at the end of training, this procedure yielded a controller and a valid Control Lyapunov function. Follow up works extended this approach to cases wherein only the linearized dynamics are known (Zhou et al., 2022), and to discrete-time dynamics as well (Wu et al., 2023). More recently, other works have addressed the challenge of verifying the learned Lyapunov function, with (Dai et al., 2021) using integer programming for verification, and (Yang et al., 2024) using the $\alpha - \beta$ CROWN verifier for output feedback control with neural networks. More recent work applied this approach to certifying stability of Reinforcement Learning policies (Long et al., 2025). Other stability certificates have also been parameterized by neural networks - contraction metrics (Lopez & Slotine, 2020) were also parameterized with neural networks (Sun et al., 2021). Neural certificates have also been applied to safety certification in works such as (Liu et al., 2023; Dawson et al., 2023). In contrast, our work is the first to synthesis controllers using neural density functions, which also provides the added benefit of being able to blend the controllers smoothly.

**Lyapunov Density Functions for Stability and Control**   Density functions for stability were first proposed in (Rantzer, 2001), where they were stated as a 'dual' to Lyapunov functions. More recent analysis studies density functions for certifying instability as well (Furtat & Gushchin, 2021; Furtat, 2025). The density functions possess a remarkable convexity property that facilitates control synthesis using sum-of-squares (SOS) programming (Prajna et al., 2004), which has subsequently been used for data-driven control (Dai & Sznaier, 2020; Huang et al., 2025). Almost-everywhere regions of attraction certified by density functions have also been studied (Sinha et al., 2025; Masubuchi, 2007; Masubuchi & Kikuchi, 2021). Subsequent

work showed that density functions could be used to 'blend' controllers together in (Rantzer & Ceragioli, 2001), and was extended to safety and reachability analysis in (Prajna & Rantzer, 2005; Prajna, 2003; Prajna & Rantzer, 2007; Zheng et al., 2023). Other work connected density functions to the Perron-Frobenius operator (Vaidya & Mehta, 2008; Vaidya et al., 2010; Vaidya & Chinde, 2015), using what they called Lyapunov measures. More recent work have studied density functions as solutions to the Liouville PDE (Chen & Ames, 2019; Chen et al., 2020; Meng et al., 2022), and have applied density functions to navigation tasks (Moyalan et al., 2024; Deka et al., 2023; Zheng et al., 2023). Our work differs from these as the first to parameterize density functions using neural networks, with a view toward blending different controllers. Our work is also the first to characterize the regions of attraction of blended controllers using density functions.

## 3. Background

In this section, we discuss prior work and state foundational definitions necessary for our main results, along with the notation used.

**Notation**   In this section, we introduce the notation used in this work. Let $[n] = \{1, \cdots, n\}$ for any integer $n$. Let $\nabla \cdot f(x) = \sum_i \frac{\partial f(x)}{\partial x_i}$ denote the **divergence** of a vector field $f(x)$. Similarly, let $L_f g(x) = \nabla g(x)^\top f(x)$. Let $\mathbf{1}_C[x]$ denote the indicator function for a set $C \subset \mathbb{R}^d$. For a dynamical system $\dot{x} = f(x)$, let $\phi(t, x_0)$ denote the trajectory at time $t$ from initial condition $x_0$. For a Borel set $A$, let $m(A)$ denote its Lebesgue measure Let $C^k$ denote the space of functions that are at least k-times differentiable, such that all $l \leq k$-th derivatives are continuous. Let $\wedge$ and $\vee$ denote the logical *AND* and *OR* operators, respectively. Note that an extended class $\mathcal{K}_\infty$ function is a function $\alpha : \mathbb{R} \to \mathbb{R}$ that is strictly increasing and with $\alpha(0) = 0$.

**Dynamical Systems and Stability**   We consider autonomous dynamical systems of the form

$$\dot{x} = f(x) \tag{1}$$

where $f : S \to \mathbb{R}^n$ is a Lipschitz continuous vector field, and $S \subseteq \mathbb{R}^n$ with $0 \in S$. Without loss of generality, we assume that the equilibrium is at the origin [2]. The stability and almost everywhere (a.e.) stability of the system 1 is defined as follows.

**Definition 3.1** (Asymptotic Stability (Khalil))**.** We say that system of (1) is stable at origin if for any $\epsilon \in \mathbb{R}^+$, there exists $\delta_\epsilon \in \mathbb{R}^+$ such that if $\|x(0)\| < \delta$ then $\|x(t)\| < \epsilon$ for

[2]Unless mentioned explicitly, we assume 0 to be equilibrium throughout this paper.

all $t \geq 0$. The system is asymptotically stable at the origin if it is stable and also

$$\lim_{t \to \infty} \|x(t)\| = 0 \text{ for all } \|x(0)\| < \delta. \tag{2}$$

We next define *almost-everywhere* stability.

**Definition 3.2** (Almost-everywhere stability)**.**  Consider the system (1) with equilibrium $0 \in S$, where $S \subseteq \mathbb{R}^n$. We say that the origin is an *almost-everywhere stable* equilibrium of (1) if

$$m\left(\left\{x \in S : \lim_{t \to \infty} \phi(t, x) \neq 0\right\}\right) = 0,$$

**Definition 3.3** (Region of Attraction)**.**  Let $\phi(t, x)$ be the solution of (1) with equilibrium 0. The region of attraction(RoA) is defined as the largest set $Q$ of points $x$ such that $\lim_{t \to \infty} \phi(t, x) = 0$ and $\phi(t, x) \in Q$ for all $t$.

Similar to Definition 3.2, we define the almost-everywhere region of attraction.

**Definition 3.4** (Almost-everywhere Region of Attraction)**.**  Consider the system (1) with equilibrium at $0 \in S$. The almost-everywhere region of attraction of 0, denoted $Q$, is the largest measurable set $Q \subseteq S$ containing $x^\star$ such that

$$m(\{x \in D : \lim_{t \to \infty} \phi(t, x) \neq 0\} \cup .$$
$$\{x \in Q : \exists t \geq 0 \text{ such that } \phi(t, x) \notin Q\}) = 0.$$

The *almost everywhere* region of attraction is the largest set $Q$ containing $x^\star$ such that $\lim_{t \to \infty} \phi(t, x) = x^*$ and $\phi(t, x) \in D$ for almost all initial conditions $x \in Q$.

**Lyapunov Functions for Stability Analysis**   In this section, we provide a brief review of Lyapunov's method for certifying stability of (1), which requires finding a function $V : \mathbb{R}^n \to \mathbb{R}$, where $V(0) = 0$ and $V(x) > 0$ for all $x \in \mathbb{R}^n \backslash \{0\}$, that satisfies

$$\dot{V}(x) = \nabla V(x)^\top f(x) \leq 0. \tag{3}$$

This method is extended to control synthesis of systems of the form

$$\dot{x} = f(x) + g(x)u(x) \tag{4}$$

where $f : S \to \mathbb{R}^n, g : S \to \mathbb{R}^{n \times m}$ is a Lipschitz continuous vector field, $S \subseteq \mathbb{R}^n$ with $0 \in S$ and the feedback control is defined by a continuous function $u : S \to \mathbb{R}^m$. Applying Lyapunov's method to (4), (3) reduces to,

$$\dot{V}(x) = \nabla V(x)^\top (f(x) + g(x)u(x)) \leq 0 \tag{5}$$

**Rantzer's Density Functions for Stability Analysis** An alternative stability criterion was proposed in (Rantzer, 2001) which certifies almost everywhere stability, a weaker notion but enables stability analysis of equlibiria even in the presence of unstable equilibria or saddle point. We discuss extensions of this method to certifying instability in Appendix C.

**Theorem 3.5** ((Rantzer, 2001)). *Given the system 1, where $f \in C^1(\mathbb{R}^n, \mathbb{R}^n)$ and $f(0) = 0$, suppose there exists a non-negative density function $\rho \in C^1(\mathbb{R}^n \backslash \{0\}, \mathbb{R})$ such that $\rho(x)f(x)/\|x\|$ is integrable on $\{x \in \mathbb{R}^n : \|x\| \geq 1\}$ and*

$$\nabla \cdot [\rho(x)f(x)] > 0 \quad \text{for almost all } x \quad (6)$$

*Then, for almost all initial states $x(0)$ the trajectory $x(t)$ exists for $t \in [0, \infty)$ and tends to 0 as $t \to \infty$. Moreover, if the equilibrium $x = 0$ is stable, the conclusion remains valid even if $\rho$ takes negative values.*

It is worth noting that the density function $\rho(x)$ can also be used to derive a Lyapunov function $V$ for the given system (1). Specifically, when the divergence condition $\nabla \cdot f(x) \leq 0$ holds, we may define $V(x) = \rho(x)^{-1}$ as presented in (Rantzer, 2001, Proposition 2).

**Control Synthesis with Density Functions** Consider a dynamical system described by (4). Assume there exists a $\rho$ as described in Theorem 3.5 such that:

$$\nabla \cdot \rho(x)(f(x) + g(x)u(x)) > 0 \quad \text{for almost all x} \quad (7)$$

holds true. The divergence criterion (7) is linear in the pair $(\rho, \rho u)$; this enable convex control synthesis by defining $\rho(x)u(x) = \psi(x)$, and the divergence criterion reduces to:

$$\nabla \cdot (\rho(x)f(x) + \psi(x)g(x)) > 0 \quad \text{for almost all } x \quad (8)$$

Thus, the control synthesis problem reduces to finding a pair of functions $(\rho, \psi)$ that satisfies (8). Furthermore, as noted in (Rantzer & Ceragioli, 2001), the linearity of the divergence condition (6) enables the smooth blending of nonlinear controllers by linearly combining $N$ functions $\{\rho_i, \psi_i\}_{i=1}^N$. In Section 4, we define the notion of blending and analyze the extent to which the RoAs of smoothly blended controllers improve upon those of the constituent controllers.

**Control Barrier Functions for Safe Control Synthesis**
We consider a set $\mathcal{C}$ defined as the superlevel set of a continuously differentiable function $h : S \subset \mathbb{R}^n \to \mathbb{R}$, yielding:

$$\mathcal{C} = \{x \in S \subset \mathbb{R}^n : h(x) \geq 0\}$$
$$\partial C = \{x \in S \subset \mathbb{R}^n : h(x) = 0\}$$
$$\text{Int}(\mathcal{C}) = \{x \in S \subset \mathbb{R}^n : h(x) > 0\} \quad (9)$$

We refer to $\mathcal{C}$ as the safe set.

**Definition 3.6** (Safe). The set $\mathcal{C}$ is **forward invariant** if for every $x_0 \in \mathcal{C}, x(t) \in \mathcal{C}$ for $x(0) = x_0$ and all $t \geq 0$ The system (4) is **safe** with respect to the set $\mathcal{C}$ if the set $C$ is forward invariant.

Safety of a system under a control $u$ can be certified using control barrier function (CBF) as described in (Ames et al., 2019).

**Definition 3.7** (Control Barrier Function). Suppose $\mathcal{C} \subset S \subset \mathbb{R}^n$ be defined as in (9), then $h$ is a control barrier function (CBF) if there exists an extended class $\mathcal{K}_\infty$ function $\alpha$ such that for the control system described by Equation (4):

$$\sup_{u \in U}[L_f h(x) + L_g h(x)u(x)] \geq -\alpha(h(x)) \, \forall x \in S \quad (10)$$

## 4. Regions of Attraction of Smoothly Blended Controllers

In this section, we address the challenge of quantifying the extent to which blending controllers improves the region of attraction of the closed-loop system. We show that the a.e. region of attraction of the blended system subsumes the union of a.e. regions of attraction of the constituent dynamics.

### 4.1. Certifying Almost-Everywhere Forward Invariant Sets with Density Functions

In this section, we consider the problem of certifying almost-everywhere forward invariance of sets using density functions. We will show, in Theorem 4.1, that under mild assumptions, the superlevel sets of density functions serve as inner approximations of the a.e. region of attraction.

**Theorem 4.1.** *Consider the system (1), and assume it is almost-everywhere stable, certified by density function $\rho(x)$ with bounded and connected superlevel sets. Then, the set $D := \{x \in \mathbb{R}^n : \rho(x) > \tau\}$ is forward invariant if and only if $\nabla \cdot [f\rho](x) > \tau \nabla \cdot f(x)$ for all $x \in \partial D$.*

*Proof Sketch.* We prove sufficiency by contradiction, by showing that if any trajectories escape the superlevel set, then the divergence condition's positivity yields a contradiction. Necessity is proved by using the fact that on invariant sets, $\rho$ must have a positive time derivative at the boundary. The complete proof is given in Appendix B.1. $\square$

*Remark* 4.2. In practice, we enforce the condition $\nabla \cdot [f\rho](x) \geq \tau \nabla \cdot [f](x)$ for all $x$. While this approach is more conservative, it is also a constraint that is easier to enforce.

If the system (1) has nonpositive divergence - that is, $\nabla \cdot f(x) \leq 0$, then, by (Rantzer, 2001, Proposition 2), we can construct a Lyapunov function from the density function,

which guarantees that the density function's superlevel sets are inner approximations of the true or region of attraction. We state this in Corollary 4.3.

**Corollary 4.3.** *Consider the system* (1)*, and assume it is stable with* $\nabla \cdot [f](x) \leq 0$*, certified by density function* $\rho(x)$ *with bounded and connected superlevel sets. Then, the set* $D := \{x \in \mathbb{R}^n : \rho(x) > \tau\}$ *is forward invariant.*

*Proof.* If $\nabla \cdot [f](x) \leq 0$, it follows that for any $x$, $d\rho(x)/dt = \nabla\rho(x)^\top f(x) > \rho|\nabla \cdot [f](x)| \geq 0$. Thus, for any $\tau$ and the associated set $D := \{x : \rho(x) \geq \tau\}$, $d/dt(\rho(x(t)) - \tau) \geq 0$ for any $x(t) \in \partial D$, and thus, by Theorem 4.1, $D$ is forward invariant. $\square$

## 4.2. Smoothly Blending Controllers using Density Functions

In this section, we formalize the problem of smoothly blending controllers using density functions. We begin by defining the constituent systems and the blended control system. Suppose we have $N$ systems governed by the following differential equations:

$$\dot{x} = F_i(x) = f(x) + g(x)u_i(x), \quad i \in [N]. \quad (11)$$

Suppose we have density functions $\rho_i$ that certify the a.e. stability of the system $F_i$. Let $Q_i$ denote the region of attraction of $F_i(x)$, and define

$$D_i := \{x \in S : \rho_i(x) \geq \tau_i\} \subset Q_i. \quad (12)$$

Then, define

$$\rho^\star(x) = \sum_i \rho_i(x), \ u^\star(x) = \sum_i \frac{\rho_i(x)u_i(x)}{\rho^\star(x)},$$
$$F^\star(x) = f(x) + g(x)u^\star(x), \quad (13)$$

and let $Q^\star$ be the almost everywhere region of attraction of $F^\star$. We say $u^\star$ is the *blended controller*, $\rho^\star$ is the *blended density*, and $F^\star$ is the *blended system*.

## 4.3. Improved Almost-Everywhere Regions of Attraction with Blended Controllers

In this section, we show that given a set of controllers that create closed loop systems whose a.e. stability is certified by density functions, the a.e. region of attraction obtained by blending the individual controllers is a superset of the union of the individual a.e. regions of attraction.

**Theorem 4.4.** *Suppose we have vector fields* $F_1, \cdots, F_N$ *as defined in* (11)*, and suppose* $\rho_i$ *are density functions certifying their a.e. stability. Let* $D_i \subset S$ *be defined as in* (12)*, and suppose* $\tau(x) = \min_i \tau_i$*. Let* $F^\star$*,* $\rho^\star$ *and* $u^\star$ *be defined as in* (13)*, let* $Q^\star$ *be the almost-everywhere region of attraction of* $F^\star(x)$ *contained in* $S$*. Define* $D^\star := \{x \in$

$S : \rho^\star(x) \geq \tau\}$*, where* $S$ *is a compact set containing the origin, and* $\tau = \min_i \tau_i$*. Then,*

$$\bigcup_i D_i \subseteq D^\star \subseteq Q^\star.$$

*Moreover, if we assume* $m(Q_i \backslash D_i) = 0$*, then* $m(Q^\star) \geq m\left(\bigcup_i Q_i\right)$.

*Proof Sketch.* We first show that the set $\cup_i D_i$ is a subset of the certified and true regions of attraction $D^\star$ and $Q^\star$ of the blended closed loop system, and then show that the measure of the symmetric difference set between $A$ and the union of $Q_i$ is 0, to show that the union of $Q_i$ is a subset of $Q^\star$. The complete proof is presented in Appendix B.2. $\square$

Theorem 4.4 shows that the blended controller achieves a better region of attraction than *all* of the constituent controllers combined; this is the first result of it's kind to showcase this.

*Remark* 4.5. As noted in (Rantzer, 2001, Proposition 2), if the divergence of the closed loop vector field is negative, then a Lyapunov function can be constructed using the density function. However, even in the setting wherein for each $i$, $\nabla \cdot [F_i(x)] \leq 0$, one cannot guarantee that $\nabla \cdot [F^\star(x)] \leq 0$ (where $F^\star(x)$ is defined as in (13)) as well. Thus, even if each of the individual controllers $u_i(x)$ is globally asymptotically stabilizing, we can only conclude that the blended controller $u^\star(x)$ yields a closed loop system that is almost-everywhere asymptotically stable.

# 5. Synthesizing Neural CDFs

In this section, we address the problem of stability analysis and control synthesis with neural density functions, which use a novel parameterization that uses pairs of networks, and guarantees integrability as required in Theorem 3.5.

## 5.1. Certifying Stability with Neural Density Functions

We begin by studying the problem of stability analysis with neural density functions. Suppose we are given the system (1), where $f(x)/\|x\|$ is integrable on the set $\{x \in \mathbb{R}^n : \|x\| \geq 1\}$. To analyze the almost-everywhere stability of this system, we solve the following feasibility problem.

$$\text{Find} \ \ \rho_\theta \in C^1(\mathbb{R}^n \backslash \{0\}, \mathbb{R}_{\geq 0})$$
$$\text{s.t.} \ (6) \text{ holds, and} \int_{x : \|x\| > 1} \rho_\theta(x) < \infty \quad (14)$$

Here, $\rho_\theta$ is a parametric density function with parameters $\theta$, wherein we train the parametric model on a finite dataset, and use a verifier to generate counterexamples; when the verifier returns UNSAT, we terminate the process and obtain a stability certificate.

The goal in this section is to learn $\rho_\theta(x)$ using a neural network, in the CEGIS framework. To do so, we define the Neural Density Function Loss (NDF Loss):

$$L_\zeta(\theta) = \mathbb{E}_{x \sim \zeta(S)}[\max(0, -\rho_\theta(x))$$
$$+ \max(0, -\nabla \cdot \rho_\theta(x)f(x))] \quad (15)$$

where $x$ is the state, and $\zeta(S)$ is the distribution over the state space from which the training data is sampled. It is straightforward to see that if $L_\zeta(\theta) = 0$, the feasibility problem (14) has a solution, provided that $\rho_\theta(x)$ is integrable. This illuminates a key challenge in learning neural density functions - that not all parametrizations yield networks that are integrable. Thus, we require a novel parametrization for $\rho_\theta(x)$ that guarantees integrability, and whose feasibility is sufficient to prove a.e. stability. We discuss this parameterization in the sequel in the context of control synthesis.

## 5.2. Learning to Control with Neural Control Density Functions

We address the challenge of control synthesis for systems such as (4) by finding pairs of parametric functions $(\rho_\theta, \psi_\theta)$, and setting $u_\theta(x) = \rho_\theta(x)/\psi_\theta(x)$.

We learn $(\rho, \psi)$ using the Neural Control Density Loss

$$L_\zeta(\theta) = \mathbb{E}_{x \sim \zeta(S)}[\max(0, -\rho_\theta(x))$$
$$+ \max(0, -\nabla \cdot [\rho_\theta f + g\psi_\theta](x))] \quad (16)$$

where $\zeta(S)$ is a distribution over states $x \in S$ from which data is drawn.

The $(\rho_{\theta^\star}, u_{\theta^\star})$ which certifies the stability of the system Equation (4) is a feasible solution if $L_\zeta(\theta^\star) = 0$, provided that $\rho_{\theta^\star}$ is integrable. We utilize the same approach to learning to control as we do with (15) We can also define density functions from which Lyapunov functions can be recovered by adding a regularization term encouraging negative divergence of the dynamics to (16) as follows:

$$L'_\zeta(\theta) = L_\zeta(\theta) + \mathbb{E}_{x \sim \zeta(S)}[\max(0, \nabla \cdot [f + gu](x))]$$
$$(17)$$

where $L_\zeta(\theta)$ is the density risk described in (16).

As noted in Section 4.1, we cannot directly apply the method proposed in (Chang et al., 2019; Zhou et al., 2022), as the neural density functions must satisfy the integrability condition stated in (Rantzer, 2001). While standard gradient-based optimizers are capable of finding pairs $(\rho, u)$ achieving 0 train error, they cannot enforce the required integrability constraint. Inspired by parametrized representation of $(\rho, \psi)$ in (Prajna et al., 2004), we consider the following parametrized representation of $(\rho, \psi)$:

$$\rho(x) = \frac{a(x)}{e^{||x||^2 + ||b(x)||^2}}, \quad \psi(x) = \frac{c(x)}{e^{||x||^2 + ||b(x)||^2}}. \quad (18)$$

where $a : \mathbb{R}^n \to \mathbb{R}, c : \mathbb{R}^n \to \mathbb{R}^m$ and $b : \mathbb{R}^n \to \mathbb{R}^d$ for any chosen dimension '$d$'. Since the exponent term is non-negative, the condition of divergence reduces to:

$$\nabla \cdot (a(x)f(x) + g(x)c(x)) -$$
$$(f(x)a(x) + g(x)c(x))^\top (2x + \nabla||b(x)||^2) > 0 \quad (19)$$

In practice, we use Equation (19) with (16). This formulation avoids numerical issues such as exploding gradients that can arise from the exponential term in the denominator during optimization. Moreover, the parametrization described by Equation (18) ensures integrability under appropriate conditions described in Proposition 5.1.

**Proposition 5.1.** *Assume $a(x), c(x)$ are $C^1(\mathbb{R}^n \backslash \{0\}, \mathbb{R})$ and $f, g \in C^1(\mathbb{R}^n, \mathbb{R}^n)$. Further, assume there exists $K_1, K_2 > 0$ such that $|a(x)|||f(x)|| \leq K_1 e^{\alpha||x||^2}$ and $|c(x)|||g(x)|| \leq K_2 e^{\beta||x||^2}$ for some $\alpha, \beta \in (0, 1)$. Given the parametrization 18 of $(\rho, \psi)$ and $\rho > 0$, $\rho(x)(f(x) + g(x)u)/|x|$ is integrable on $\{x \in \mathbb{R}^n : |x| \geq 1\}$.*

**Proof Sketch** Under the assumed growth condition, we argue $\rho f/|x|$ is bounded by $e^{(\alpha-1)||x||^2}$ and since it is integrable on $\{x \in \mathbb{R}^n : |x| \geq 1\}$, $\rho f/|x|$ is integrable too. The full proof of Propositions 5.1 is presented in Appendix B.3.

Under mild assumptions as indicated in Proposition 5.1, the proposed parametrization of $(\rho, \psi)$ satisfies the integrability conditions. In practice, we use activation functions like `tanh` or `sigmoid` which are bounded functions and assume $f, g$ are Lipschitz. Thus, the exponential growth condition is typically satisfied for models parameterized as (18). This ensures the feasibility of $(\rho, u)$ in Equation (16), and ensures the learned certificates satisfy the integrability condition.

**Maximizing the Region of Attraction** In order to synthesize controllers that aim to maximize the region of attraction, we add terms to the loss function to ensure that $\tau$ (as defined in Theorem 4.1) is minimized. We define

$$\hat{L}_\zeta(\theta) = L_\zeta(\theta) + \tau + \mathbb{E}_{x \sim \zeta(S)}[\max(0, \nabla \cdot [(f(\rho_\theta - \tau)$$
$$+ g\psi_\theta(1 - \tau/\rho_\theta)](x))] \quad (20)$$

This loss can also be simplified by setting $\tau$ a priori to a sufficiently small value.

**Falsifying Control Density Functions** Let $(\rho, \psi)$ be a candidate density function pair for a dynamical system defined in state space $S$. Let $\epsilon > 0$ be a small constant parameter that bounds the tolerable numerical error. The Density falsification constraint is the following first order logic formula over real numbers:

$$\Phi_\epsilon = \left( \sum_{i=1}^n x_i^2 \geq \epsilon \right) \wedge \quad (21)$$
$$(\rho(x) \leq 0 \ \vee \ \nabla \cdot [\rho_\theta f + g\psi_\theta](x) \leq 0)$$

where $x$ is bounded in the state space $S$ of the system.

The falsifier searches for solutions to $\Phi_\epsilon(x)$ within $S$. If such a counterexample is found, it is added to the learner's training set to refine the density and control models. We employ SMT solvers such as dReal, which has been successfully used in related verification settings (Chang et al., 2019; Zhou et al., 2022), to efficiently identify violating states. Our algorithm is stated in Algorithm 1 and provide a discussion of the algorithm in Appendix A.

---

**Algorithm 1** Neural Control Density Function

---

1: **Function** LEARNING$(X, f, g)$
2:    Initialize $a, b, c \leftarrow \text{NN}_\theta$
3: **repeat**
4:      Evaluate density risk $L_\zeta(\theta)$
5:      $\theta \leftarrow \theta - \eta\nabla_\theta L_\zeta(\theta)$
6: **until** stopping condition is met
7:    **Return** $a, b, c$
8: **Function** MAIN()
9:   **Input:** dynamical system $(f, g)$, radius $\epsilon$, precision $\delta$, initial states $X$
10: **repeat**
11:      $a, b, c \leftarrow$ LEARNING$(X, f, g)$
12:      CE $\leftarrow$ FALSIFICATION$(f, g, a, b, c, \epsilon, \delta)$
13: **until** verified
14:   Compute $\rho(x) = \dfrac{a(x)}{e^{\|x\|^2 + \|b(x)\|^2}}$
15:   Compute $u(x) = \dfrac{c(x)}{a(x)}$
16:   **Return** $\rho(x), u(x)$

---

# 6. Safe Stabilization with Neural Control Density Functions

Synthesizing controllers that are safe - that is, that satisfy a set invariance constraint - is a crucial area of research. In this section, we develop methods for synthesizing safe and stable controllers, by combining Neural Control Density Functions with control barrier functions.

## 6.1. Synthesizing Safe and Stable Controllers with Density Functions

We first propose an alternate criterion for safe-stabilization of control affine systems (4) using density functions. This formulation unifies almost everywhere stability guarantees from density function with forward invariance guarantees from CBF, as stated in Theorem 6.1.

**Theorem 6.1.** *Consider the dynamical system* (4), *and define the set* $\mathcal{C} := \{x \in \mathbb{R}^n : h(x) > 0\} \subset S$, *where* $h \in C^1(\mathbb{R}^n, \mathbb{R})$. *Suppose there exists a positive, density function* $\rho$ *and* $\psi \in C^\infty(\mathbb{R}^n, \mathbb{R})$ *such that* $f(x)\frac{\rho(x)}{\|x\|} + g(x)\frac{\psi(x)}{\|x\|}$ *is integrable on* $\{x \in \mathbb{R}^n : \|x\| \geq 1\}$, *and suppose there exists an extended class* $\mathcal{K}_\infty$ *function* $\alpha$ *such that:*

$$\nabla \cdot (\rho(x)f(x) + \psi(x)g(x)) > 0 \quad \text{for almost all } x$$
$$\rho(x)\left(L_f h + \alpha(h(x))\right) + \psi(x)L_g h(x) \geq 0 \quad (22)$$

*Then, there exists a static state feedback controller* $u(x)$

*such a solution* $\phi(t, x)$ *of* (4), *starting from almost all* $x_0 \in \mathcal{C}$, *satisfies* $\lim_{t \to \infty} \phi(t, x) = 0$ *and, for all* $x \in \mathcal{C}$ *and* $t \geq 0$, $\phi(t, x) \in \mathcal{C}$.

*Proof Sketch.* We then use Theorem 3.5, trajectories starting from almost all initial conditions in set $S$ converge to the equilibrium. Further, we show that $h$ is a control barrier function and using (Ames et al., 2019, Theorem 2) shows forward invariance of the safe set $\mathcal{C}$. The proof of Theorem 6.1 is presented in Appendix B.4. □

## 6.2. Safe Stabilization with Neural Control Density Functions

In this section, we extend the CEGIS approach for learning NCDFs for safe stabilization. We first define the Safe Density Loss, by augmenting (16) with a term that penalizes trajectories that leave the safe set,

$$SL_\zeta(\theta) = L_\zeta(\theta) + \mathbb{E}_{x \sim \zeta(S)}[\max(0, -C_h(x, \theta))] \quad (23)$$

where $L_\zeta(\theta)$ is the Control Density Risk (16) and $C_h(x, \theta) = \rho_\theta(x)(L_f h(x) + \alpha(h(x))) + \psi_\theta(x)L_g h(x)$.

We use the parametrized representation of $(\rho, \psi)$ described in Equation (18) to ensure integrability and empirical version of safe density risk[3]. We extend the density-function falsification procedure to incorporate safety violations. Specifically, counterexamples are generated whenever either the density conditions or the safety condition is violated outside a small neighbourhood of the origin. This yields the following falsification constraint:

$$\Phi_\epsilon = \left(\sum_{i=1}^n x_i^2 \geq \epsilon\right) \wedge (\rho_\theta(x) \leq 0 \ \vee$$
$$\nabla \cdot [\rho_\theta f + g\psi_\theta](x) \leq 0 \vee C_h(x, \theta) \leq 0)$$

We provide pseudocode of the algorithm in Appendix A.

# 7. Experiments

In this section, we empirically evaluate the proposed framework for learning provably stable *(NCDFs)* and demonstrate their flexibility compared to *Neural Lyapunov Controllers (NLCs)*. We organize our experiments as follows:

**(Q1) Certifying almost everywhere stability:** Can a Neural Density Function be learned to certify stability of nonlinear systems?

**(Q2) Synthesizing and Blending NCDFs:** Can density and control policies be learned jointly to certify stability? Is it

---

[3]To prioritize safety, we introduce a weighting parameter $\lambda \geq 1$ in practice that penalizes violations of the CBF constraint more heavily during training.

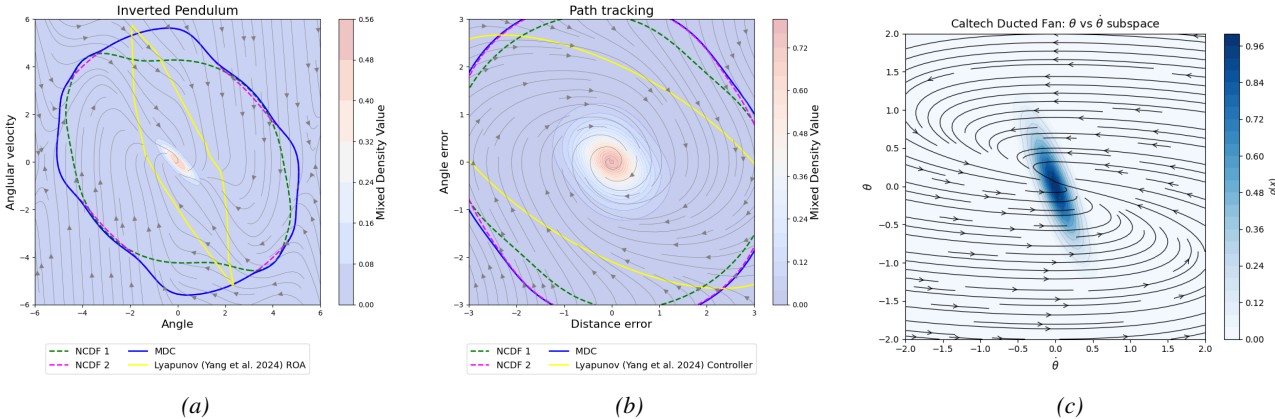

*Figure 1.* (a)-(b): Comparison of RoAs due to individual learned NCDFs, mixed NCDFs and (Yang et al., 2024) for inverted pendulum and path tracking problem respectively. (c): shows the trajectories due to the learned controller using NCDFs in $(\theta, \dot{\theta})$ subspace for caltech ducted fan system.

possible to blend these learned controllers and what is the extent to which blended controllers outperform non-blended controllers?

**(Q3) Neural Safe-Stable Control Synthesis:** Can we learn control policies which are both safe and stable using Neural Density Functions?

**Experimental Setup**  We parametrize the functions $a(x)$, $b(x)$, and $c(x)$ using feedforward neural networks distinct activation function. Specifically, $a(x)$ is modelled using `sigmoid` activations in all layers to ensure positivity, whereas $b(x)$ and $c(x)$ employ `tanh` activations in all hidden layers and a linear activation in the output layer. To avoid unintended bias in the exponential denominator of (18), we set the bias of $b(x)$ to zero.

We train the neural density controller using the empirical risk minimization (ERM) formulation of the density risk (16). For each system, we sample states uniformly from a bounded region of the state space. Datasets of size between 500 and 2000 are used for training, depending on system dimensionality and dynamics complexity. The networks are trained with learning rates in $\{10^{-2}, 10^{-4}\}$, depending on the system under consideration with early convergence. The falsifier is called every 100 steps by default, and every 1000 steps for smoother systems where violations are rare, contrasting with prior work (Chang et al., 2019), which required falsification every 10 or fewer steps.

**Certifying almost everywhere stability:**  To demonstrate the applicability of Neural Density Functions for stability analysis of nonlinear systems, we consider the following systems from (Rantzer, 2001):

$$\dot{x}_1 = -2x_1 + x_1^2 - x_2^2 \quad \dot{x}_2 = -6x_2 + 2x_1 x_2 \quad (24)$$

The system admits four equilibria: $(0,0)$, $(2,0)$, and $(3, \pm\sqrt{3})$, making the learning problem challenging. Fig-

ure 4(a) shows Neural Lyapunov method (Chang et al., 2020) fails to certify the stability of the systems while our method learns valid Neural Density Function that certifies the stability of the systems as indicated in Figure 4(b). We consider an additional system to demonstrate the use of Neural Density Function and is presented at Section D.3.

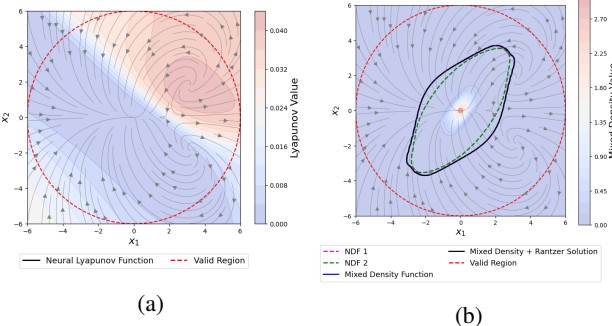

*Figure 2.* Comparison of contours due to Neural Lyapunov Function and Neural Density Function for the system (24).

**Synthesizing and Blending NCDFs:**  We extend our approach to the joint learning of control and density functions across a diverse set of benchmark systems.

- **Examples from (Prajna et al., 2004).** We learn NCDFs for the system described in Equations (33) and (34). As suggested by Figure 3a and 3b , we learn a valid controller within $\|x\| \leq 6$. The SoS-based controllers and densities $(u, \rho)$ for given systems are provided in (Prajna et al., 2004). We blend the learned NCDFs with the solutions provided in (Prajna et al., 2004) to achieve improved RoAs.

- **Inverted pendulum and path tracking.** We evaluate NCDFs on the inverted pendulum and path tracking prob-

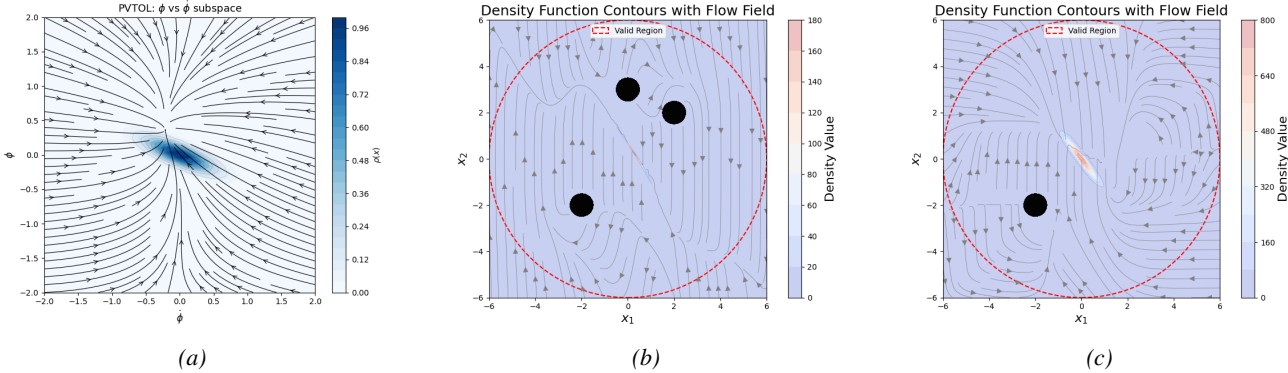

*(a)*        *(b)*        *(c)*

*Figure 3.* (a): shows the trajectories due to the learned controller using NCDFs in $(\phi, \dot{\phi})$ subspace for PVTOL system. (b)-(c): Safe-Stable control synthesis for system described by (33) and (25) respectively.

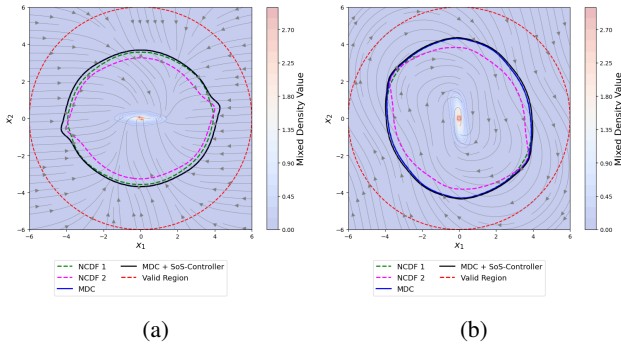

    (a)            (b)

*Figure 4.* Comparison of contours due to Neural Lyapunov Function and Neural Density Function for the system (24).

lem, both described in Section D.2, and certified regions of attraction (RoAs) against the recent approach of (Yang et al., 2024). Figure 1a and 1b shows that our method yields certified RoAs that are larger than those of (Yang et al., 2024) on the inverted pendulum and comparable on the path tracking task. For the inverted pendulum, our model learns a Neural Density Function valid within $\|x\|_2 \leq 6$ and for the path following, we learn a Neural Density Function valid within $\|x_i\| \leq 3$.

- **Caltech ducted fan and PVTOL.** We evaluate NCDFs on two higher dimensional systems: caltech ducted fan with state variables $\left[x, y, \theta, \dot{x}, \dot{y}, \dot{\theta}\right]$ and PVTOL system with state variables $\left[P_x, P_z, v_x, v_z, \phi, \dot{\phi}\right]$, whose dynamics are described in Section D.2. As shown in Figures 1c and 3a, NCDFs successfully synthesize stabilizing controllers for both systems.

**Neural Safe-Stable Control Synthesis:** We demonstrate the proposed Neural CDF-CBF framework for jointly synthesizing controllers that are both stabilizing and safe, in the sense of Theorem 6.1. We consider the dynamics described by Equation (33) with unsafe set given by $\mathcal{C}^c = \{(x_1, x_2) : (x_1 + 2)^2 + (x_2 + 2)^2 \leq 0.25\}$. We also

consider a variant of duffing oscillator dynamics:

$$\dot{x}_1 = x_2, \; \dot{x}_2 = (x_1 - 1) - (x_1 - 1)^3 - 0.1x_2 + u \quad (25)$$

with unsafe sets described by:

$$\mathcal{C}_1^c = \{(x_1, x_2) : (x_1 - 0)^2 + (x_2 - 3)^2 \leq 0.25\}$$
$$\mathcal{C}_2^c = \{(x_1, x_2) : (x_1 + 2)^2 + (x_2 + 2)^2 \leq 0.25\}$$
$$\mathcal{C}_3^c = \{(x_1, x_2) : (x_1 - 2)^2 + (x_2 - 2)^2 \leq 0.25\}$$

The safe set is defined as the complement of the union of these regions. As shown in Figures 3b and 3c, the learned controllers avoid the unsafe sets denoted by black regions and converge to the equilibrium at the origin. The learned density function and controller pair certifies the safety and stability of the system.

## 8. Conclusion

In this work, we address the problems of stability analysis and control of nonlinear systems using Neural Control Density Functions. Unlike Lyapunov-based methods, density functions certify almost everywhere stability, and facilitate smooth blending of controllers. This in turn enables us to achieve superior regions of attraction, characterized in Theorem 4.4, and allow us to certify almost-everywhere stability even in the presence of unstable equilibria. Using Neural Density Functions requires a novel parameterization stated in Proposition 5.1 to satisfy the key integrability constraint. We also extend our approach to safe stabilization. Our experiments show that regions of attraction can by improved simply by blending controllers, and that we can achieve safe stabilization as well.

**Limitations** The use of density function implies a.e. stability, meaning that a set of initial conditions with Lebesgue measure zero may fail to be certified. Moreover, the proposed parametrization for satisfying the integrability condition requires the joint optimization of three neural networks, which may extend the time required for falsification.

## Acknowledgments

Chiranjib Bhattacharyya was supported by a generous grant from HP Inc. Sahil Chaudhary gratefully acknowledges the Dr. Ubrani and Lakshmi Venkataram Travel Grant from the Office of Development and Alumni Affairs (ODAA) at the Indian Institute of Science (IISc). The authors also thank Prof. Shishir Kolathaya of IISc for insightful discussions.

## Impact Statement

This paper presents work whose goal is to advance the field of Machine Learning. There are many potential societal consequences of our work, none of which we feel must be specifically highlighted here.

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

## Appendix Outline

In this Appendix, we provide the following information.

- In Appendix A, we state the pseudocode for our Safe Neural Control Density Algorithm, and provide a brief discussion.

- In Appendix B, we provide the proofs omitted in the main paper.

- In Appendix D, we provide further evaluations of the Neural Control Density Function method.

## A. Algorithm

In this Appendix, we restate the pseudocode for our main Algorithm for synthesizing Neural Control Density Functions.

---

**Algorithm 2** Neural Control Density Function

---

1: **Function** LEARNING$(X, f, g)$
2:     Initialize $a, b, c \leftarrow NN_\theta$
3: **repeat**
4:         Evaluate density risk $L_\zeta(\theta)$
5:         $\theta \leftarrow \theta - \eta \nabla_\theta L_\zeta(\theta)$
6: **until** stopping condition is met
7:     **Return** $a, b, c$
8: **Function** MAIN()
9:     **Input:** dynamical system $(f, g)$, radius $\epsilon$, precision $\delta$, initial states $X$
10: **repeat**
11:         $a, b, c \leftarrow$ LEARNING$(X, f, g)$
12:         CE $\leftarrow$ FALSIFICATION$(f, g, a, b, c, \epsilon, \delta)$
13: **until** verified
14:     Compute $\rho(x) = \dfrac{a(x)}{e^{\|x\|^2 + \|b(x)\|^2}}$
15:     Compute $u(x) = \dfrac{c(x)}{a(x)}$
16:     **Return** $\rho(x), u(x)$

---

For joint learning of the control-density pair, we first evaluate the density risk given by (16) and optimize with respect to parameters $\theta$. After every optimization step, we evaluate if the parameters updated are valid using an SMT solver (`dreal` (Gao et al., 2013) ). The *FALSIFICATION* takes the networks, radius and precision and returns UNSAT if the density-control pair is valid, and generates counterexamples if not. Similarly, for safe control synthesis, we evaluate safe density risk given

---

**Algorithm 3** Safe Neural Control Density Function

---

1: **Function** LEARNING$(X, f, g, h)$
2:     Initialize $a, b, c \leftarrow NN_\theta$
3: **repeat**
4:         Evaluate safe density risk $SL_\zeta(\theta)$
5:         $\theta \leftarrow \theta - \eta \nabla_\theta SL_\zeta(\theta)$
6: **until** stopping condition is met
7:     **Return** $a, b, c$
8: **Function** MAIN()
9:     **Input:** dynamical system $(f, g)$, radius $\epsilon$, precision $\delta$, initial states $X$
10: **repeat**
11:         $a, b, c \leftarrow$ LEARNING$(X, f, g)$
12:         CE $\leftarrow$ FALSIFICATION$(f, g, a, b, c, h, \epsilon, \delta)$
13: **until** verified
14:     Compute $\rho(x) = \dfrac{a(x)}{e^{\|x\|^2 + \|b(x)\|^2}}$
15:     Compute $u(x) = \dfrac{c(x)}{a(x)}$
16:     **Return** $\rho(x), u(x)$

---

by Equation (23). The algorithm requires description of unsafe sets given by $h$ to evaluate the risk. The *FALSIFICATION* takes the networks, $h$, radius and precision and returns if any trajectory enters the unsafe set and validity of density-control pair. If the pair is not valid, it generates counterexamples which get added to the current dataset.

## B. Proofs of Main Results

In this Appendix, we provide proofs of the theoretical results provided in this paper.

## B.1. Proof of Theorem 4.1

Theorem 4.1 In this section, we prove Theorem 4.1, wherein we establish that the superlevel sets of density functions are forward invariant under mild assumptions.

**Theorem.** *Consider the system* (1)*, and assume it is almost-everywhere stable, certified by density function $\rho(x)$ with bounded and connected superlevel sets. Then, the set $D := \{x \in \mathbb{R}^n : \rho(x) > \tau\}$ is forward invariant if and only if $\nabla \cdot [f\rho](x) > \tau \nabla \cdot f(x)$ for all $x \in \partial D$.*

*Proof.* **Proof of sufficiency:**

Our proof for sufficiency is similar to that proposed in (Masubuchi, 2007).

First, consider the flow map of (1), $\phi(x,t)$, which is the state at time $t$ on a trajectory with initial condition $x$. Define $J(x,t) = \det(\partial\phi(x,t)/\partial x)$ to be the determinant of the Jacobian of $\phi(x,t)$. Note that $J(x,t) > 0$, as noted in (Masubuchi, 2007). Also, denote by $\partial D$ the boundary of $D$.

Next, consider a point $x \in D$, and suppose trajectories starting from this point reach $\partial D$ at time $t_\star$. Then, define

$$q(x,t) = (\rho(x) - \tau)J(x,t).$$

It follows that $q(x,0) = (\rho(x) - \tau) > 0$ (as $J(x,0) = 1$), and $q(\phi(x,t_\star),t_\star) = (\rho(\phi(x,t_\star)) - \tau)J(\phi(x,t_\star)) = 0$.

From (Masubuchi, 2007), we have

$$\dot{q}(x) = \nabla \cdot [f(\rho - \tau)](x)J(x,t) > 0$$

Thus, $q$ is a non-decreasing function. Thus, $q(\phi(x,t_\star),t_\star) > q(x,0)$. However, this contradicts the fact that $q(\phi(x,t_\star),t_\star) = 0$. Hence, trajectories cannot ever reach the boundary of $D$, and thus cannot escape $D$, thereby showing $D$ is forward invariant.

**Proof of necessity:**

Suppose $D$ is positively invariant. Define $h(t) = \rho(x(t)) - \tau$. Then, for any $t$, $x(t) \in \partial D$, $h(t) = 0$. Suppose we have $x_0 \in \partial D$. Then, $h(0) = 0$, and by the forward invariance of $D$, we have $h(t) \geq 0$ for all $t \geq 0$. Thus, it follows that $\dot{h}(0) \geq 0$. Thus,

$$\nabla\rho(x_0)^\top f(x_0) = \nabla \cdot [\rho f](x_0) - \tau\nabla[f](x_0) \geq 0.$$

Hence, necessity is proved since the condition holds for any $x_0 \in \partial D$. □

## B.2. Proof of Theorem 4.4

In this section we provide the proof for Theorem 4.4

**Theorem.** *Suppose we have vector fields $F_1, \cdots, F_N$ as defined in (11), and suppose $\rho_i$ are density functions certifying their a.e. stability. Let $D_i \subset S$ be defined as in (12), and suppose $\tau_i(x) = \tau$ for all $i$ and assume $m(Q_i \backslash D_i) = 0$. Define $\rho^\star$ as in (13), and define $A = \bigcup_i D_i$. Let $Q^\star$ be the almost everywhere region of attraction for $F^\star(x)$. Then,*

$$\bigcup_i D_i \subseteq D^\star \subseteq Q^\star \quad and \quad m(Q^\star) \geq m\left(\bigcup_i Q_i\right).$$

*Proof.* By Theorem 3.5, it follows that the set $\{x \in S : \rho^\star(x) \geq 1/r\}$ is forward invariant.
Next, choose $i$, and suppose $x' \in D_i$. Then, $\rho_i(x') \geq 0$. Next, we have

$$\rho^\star(x') = \sum_i \rho_i(x') \geq \tau_i.$$

Therefore, we have $x \in D_i^\star := \{x \in S : \rho^\star(x) \geq \tau_i\}$. From this, we have $D_i \subseteq D_i^\star$. By the forward invariance of $D^\star$, we also have $D_i^\star \subset D^\star$, since $\tau_i \geq \tau$. Then, since this inclusion holds for all $i$, it follows that

$$\bigcup_i D_i \subseteq D^\star.$$

Next, we show that $A$ is a subset of $Q^\star$. Suppose $x \in A$. Then, for some $j$, $x \in D_j$, and we have

$$\nabla \cdot [\rho^\star(x)F^\star(x)] = \sum_i \nabla \cdot [\rho_i(x)F_i(x)] \geq \nabla \cdot [\rho_j(x)F_j(x)] \geq \tau > 0.$$

Thus, $A \subseteq D^\star \subseteq Q^\star$, since the divergence of the blended system is always positive on $A$, and from Theorem 3.5, we have $m(A \backslash Q^\star) = 0$. Next, let $U = \bigcup_i Q_i$. We want to bound the measure of the symmetric difference between $A$ and $U$; that is, $(A \backslash U) \cup (U \backslash A)$. By the properties of symmetric difference, we have

$$(A \backslash U) \cup (U \backslash A) \subseteq \bigcup_i (D_i \backslash Q_i) \cup (Q_i \backslash D_i),$$

from which it follows that
$$m((A \backslash U) \cup (U \backslash A)) = 0.$$

Next, we need to show that $U$ is a subset of $Q^\star$. To do so, note that $U \backslash Q^\star \equiv (U \backslash A) \cup (A \backslash Q^\star)$. Then, we have

$$m(U \backslash Q^\star) = m((U \backslash A) \cup (A \backslash Q^\star)) \leq m(U \backslash A) + m(A \backslash Q^\star) = 0.$$

Thus, $U \subseteq Q^\star$, and thus, $m(Q^\star) \geq m(\cup_i Q_i)$. $\qquad\square$

## B.3. Proof of Proposition 5.1

In this section, we provide the proof for Proposition 5.1

**Proposition.** *Assume $a(x), c(x)$ are $C^1(\mathbb{R}^n \backslash \{0\}, \mathbb{R})$ and $f, g \in C^1(\mathbb{R}^n, \mathbb{R}^n)$. Further, assume there exists $K_1, K_2 > 0$ such that $|a(x)| \| f(x) \| \leq K_1 e^{\alpha \|x\|^2}$ and $|c(x)| \| g(x) \| \leq K_2 e^{\beta \|x\|^2}$ for some $\alpha, \beta \in (0, 1)$. Given the parametrization 18 of $(\rho, \psi)$ and if $\rho > 0$, $\rho(x)(f(x) + g(x)u)/|x|$ is integrable on $\{x \in \mathbb{R}^n : |x| \geq 1\}$.*

*Proof.* It suffices to show $\frac{a(x)f(x)}{\|x\| e^{\|x\|^2 + \|b(x)\|^2}}$ is integrable over the set $S = \{x \in \mathbb{R}^n : |x| \geq 1\}$. Observe that, for $x \in S$

$$\left\| \frac{a(x)f(x)}{\|x\| e^{\|x\|^2 + \|b(x)\|^2}} \right\| \leq \frac{|a(x)| \| f(x) \|}{e^{\|x\|^2 + \|b(x)\|^2}} \leq \left\| \frac{a(x)f(x)}{e^{\|x\|^2}} \right\|$$
$$\leq K_1 e^{(\alpha-1)\|x\|^2}$$

Since, $\alpha - 1 < 0$ and $K_1 e^{(\alpha-1)\|x\|^2}$ is integrable, so $\rho f/|x|$ is integrable on S.

An analogous argument applies to $\rho(x)g(x)u(x)/|x|$, completing the proof. $\qquad\square$

## B.4. Proof of Theorem 6.1

In this section, we provide the proof for Theorem 6.1.

**Theorem.** *Let $S$ be a bounded set such that $0 \in S$. Consider the dynamical system (4), and define the set $C := \{x \in S : h(x) > 0\} \subset S$, where $h \in C^1(\mathbb{R}^n, \mathbb{R})$. Suppose there exists a positive, density function $\rho$ and $\psi \in C^\infty(\mathbb{R}^n, \mathbb{R})$ such that $f(x)\frac{\rho(x)}{\|x\|} + g(x)\frac{\psi(x)}{\|x\|}$ is integrable on $\{x \in \mathbb{R}^n : |x| \geq 1\}$, and suppose there exists an extended class $\mathcal{K}_\infty$ function $\alpha$ such that:*

$$\nabla \cdot (\rho(x)f(x) + \psi(x)g(x)) > 0 \quad \text{for almost all } x \in S$$
$$\rho(x)(L_f h(x) + \alpha(h(x))) + \psi(x)L_g h(x) \geq 0 \quad \text{for all } x \in S \cap C. \tag{26}$$

*Then, there exists a static state feedback controller $u(x)$ such a solution $\phi(t, x)$ of (4), starting from almost all $x_0 \in C$, satisfies $\lim_{t \to \infty} \phi(t, x) = 0$ and, for all $x \in C$ and $t \geq 0$, $\phi(t, x) \in C$.*

*Proof.* Note that $f(x)\frac{\rho(x)}{\|x\|} + g(x)\frac{\psi(x)}{\|x\|}$ is integrable on $\{x \in \mathbb{R}^n : |x| \geq 1\}$ and substituting $\psi(x) = \rho(x)u(x)$ reduces the divergence criterion to:

$$\nabla \cdot (\rho(x)(f(x) + g(x)u(x)))$$

Thus, by Theorem 3.5, almost all initial states $x(0)$, the trajectory $x(t)$ exists for $t \in [0, \infty)$ and tends to $0$ as $t \to \infty$, i.e. for almost all $x(0) \in C \subset S$,

$$\lim_{t \to \infty} x(t) = 0$$

Define the state feedback controller $u(x) = \psi(x)/\rho(x)$. If there exists an extended class $\mathcal{K}_\infty$ function $\alpha$ such that:

$$\rho(x)\left(L_f h(x) + \alpha(h(x))\right) + \psi(x)L_g h(x) \geq 0$$
$$L_f h(x) + \alpha(h(x)) + \frac{\psi(x)}{\rho(x)}L_g h(x) \geq 0 \quad \text{(since $\rho$ is a non-negative function)}$$
$$L_f h(x) + u(x)L_g h(x) \geq -\alpha(h(x))$$

Thus, $h$ is a control barrier function and using (Ames et al., 2019, Theorem 2),

$$\inf_{t \geq 0} h(x(t)) > 0$$

Therefore, for almost all initial conditions $x_0 \in C$, the solution of (4) converges to the equilibrium while remaining in the safe set for all time. This completes the first part of the proof.

For the second part, we just need to prove $(\rho^*, u^*)$ satisfies the condition of CBFs as described. If $(\rho_i, \psi_i)$ satisfies the following condition:

$$\rho_i(x)(L_f h(x) + \alpha(h(x))) + \psi_i(x)L_g h(x) \geq 0 \quad \forall i \in [N]$$
$$\sum_{i=1}^{N} \rho_i(x)(L_f h(x) + \alpha(h(x))) + \psi_i(x)L_g h(x) \geq 0$$
$$\left(\sum_{i=1}^{N} \rho_i(x)\right)(L_f h(x) + \alpha(h(x))) + \left(\sum_{i=1}^{N} \psi_i(x)\right)L_g h(x) \geq 0$$
$$L_f h(x) + \alpha(h(x)) + \frac{\sum_{i=1}^{N} \psi_i(x)}{\sum_{i=1}^{N} \rho_i(x)}L_g h(x) \geq 0$$
$$L_f h(x) + u^\star(x)L_g h(x) \geq -\alpha(h(x)) \quad \text{since,} \psi_i = \rho_i u_i, u^\star = \frac{\sum_{i=1}^{N} \rho_i u_i}{\sum_{i=1}^{N} \rho_i}$$

Thus, the blended controller using density ensures the stability as well as safety of the system. $\square$

## C. Certifying Instability with Density Functions

While certifying stability of dynamical systems has long been studied, certifying instability of dynamical systems is also an important area of research (Chetaev, 1961; Furtat & Gushchin, 2021; Mohammadi & Spong, 2022). In (Furtat & Gushchin, 2021), instability results using density functions were proposed, which we state below.

**Theorem C.1.** *Suppose we have system (1), and suppose $f(0) = 0$. If there exists a nonnegative density function $\rho \in C^1(\mathbb{R}^n\backslash\{0\}, \mathbb{R})$ such that $\rho(x)f(x)/\|x\|$ is integrable on $\{x \in \mathbb{R}^n : \|x\| \geq 1\}$ and*

$$\nabla \cdot [\rho f](x) < 0 \text{ for almost all } x, \tag{27}$$

*then, for almost all $x(0)$, the trajectory $x(t)$ tends to $\infty$ as $t \to \infty$. If $\nabla \cdot [f](x) > 0$, then all trajectories $x(t) \to \infty$ as $t \to \infty$.*

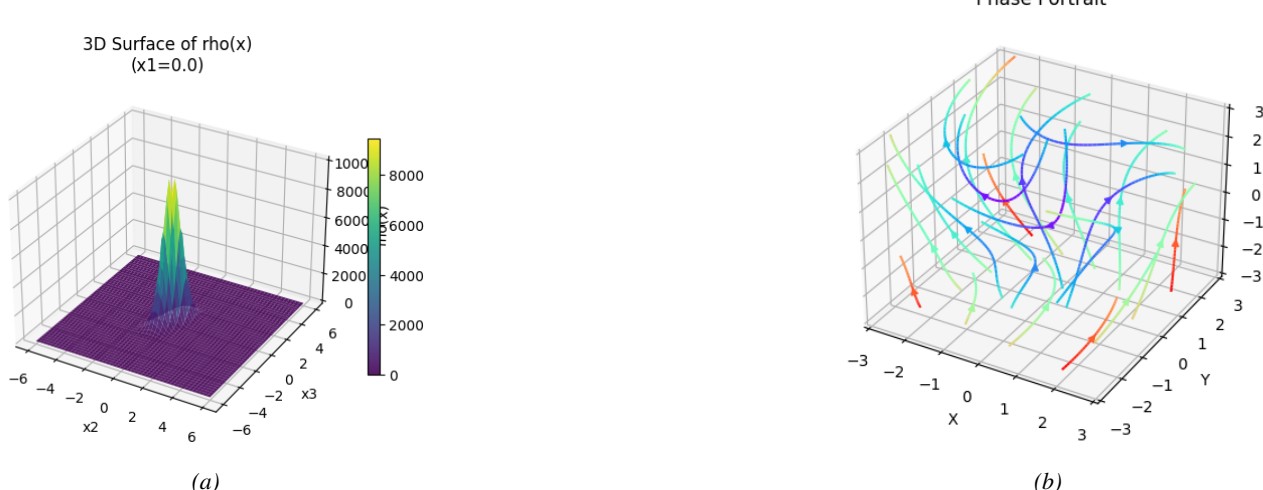

*Figure 5.* (a): valid learned density for the system, (b) 3D phase portrait of the given system

We evaluate Neural Density Functions for certifying instability. We consider the system:

$$\begin{bmatrix} \dot{x} \\ \dot{y} \\ \dot{z} \end{bmatrix} = \begin{bmatrix} x + yz \\ y - xz \\ -2z + x^2 + y^2 \end{bmatrix} \tag{28}$$

The given system admits an unstable equilibria at the origin. The analysis of such system are typically hard but density function overcome issues involved in such systems. As shown in the figure Figure 6, we learn a valid density pair which is non negative.

## D. Additional Experiments and Details

### D.1. Forward Invariance

We consider the dynamics of inverted pendulum and path tracking described by Equations (29) and (30) respectively.

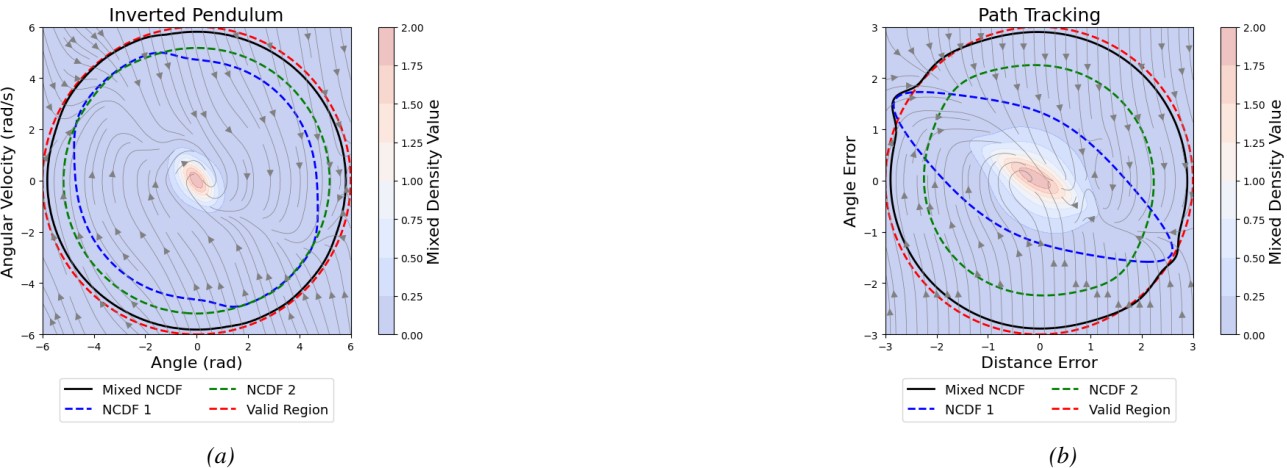

*Figure 6.* (a)-(b): Density contour for the control learned by enforcing forward invariance criterion in Equation (20) for inverted pendulum and path tracking respectively.

We learn a valid NCDF for inverted pendulum and path tracking while the forward invariance criterion is enforced.

**D.2. Dynamics of the systems**

We describe the dynamics of the system we considered in the experiment section.

**Inverted Pendulum**    We consider the classical inverted pendulum problem to evaluate our method. The system models a pendulum of mass $m$ and length $L$ with state variables given by the angular displacement $\theta$ and angular velocity $\dot{\theta}$. The control input $u$ represents the external torque applied at the pivot, and the continuous-time dynamics are given by

$$\dot{\theta} = \dot{\theta}, \quad \ddot{\theta} = \frac{MgL\sin(\theta) - \mu\dot{\theta}}{ML^2} + \frac{u}{ML^2} \tag{29}$$

where $g$ denotes gravitational acceleration and $c$ is the viscous damping coefficient.

**Path Following**    We consider the path tracking problem for a wheeled vehicle using the standard kinematic bicycle model. The system dynamics are defined in terms of the heading angle error $\theta_e$ and the lateral distance error $d_e$ to the desired path. The control input is the steering angle rate $u$, and the continuous dynamics can be written as

$$\dot{d}_e = v\sin\theta_e, \quad \dot{\theta}_e = -\frac{\cos\theta_e}{1 - d_e\theta_e} + \frac{v}{L}(\tan u) \tag{30}$$

where $v$ is the vehicle's velocity and $L$ is the wheelbase length. We assume $\tan u \approx u$ and circular path of unit radius as the reference trajectory.

**PVTOL Dynamics**    The dynamics is given by:

$$\begin{bmatrix} \dot{p}_x \\ \dot{p}_z \\ \dot{v}_x \\ \dot{v}_z \\ \dot{\phi} \\ \ddot{\phi} \end{bmatrix} = \begin{bmatrix} v_x\cos\phi - v_z\sin\phi \\ v_x\sin\phi + v_z\cos\phi \\ \dot{\phi} \\ v_z\dot{\phi} - g\sin\phi \\ -v_x\dot{\phi} - g\cos\phi \\ 0 \end{bmatrix} + \begin{bmatrix} 0 & 0 \\ 0 & 0 \\ 0 & 0 \\ 0 & 0 \\ \frac{1}{m} & \frac{1}{m} \\ \frac{l}{J} & \frac{-l}{J} \end{bmatrix} u \tag{31}$$

where $g = 9.8, m = 4, l = 0.25, J = 0.0475$.

**Caltech Ducted Fan**    The system describes the motion of a landing aircraft in hover mode with two forces $u_1$ and $u_2$. The state variables $x, y, \theta$ denote the position and orientation of the centre of the fan. The dynamics is given by:

$$\ddot{x} = \frac{1}{m}(-d\dot{x} + u_1\cos\theta - u_2\sin\theta)$$
$$\ddot{y} = \frac{1}{m}(-d\dot{y} + u_1\sin\theta + u_2\cos\theta) - g$$
$$\ddot{\theta} = \frac{r}{I}u_1$$

with $m = 11.2, r = 0.156, I = 0.0462, d = 0.1$ and $g = 0.28$ where the state variables is $\begin{bmatrix} x & y & \theta & \dot{x} & \dot{y} & \dot{\theta} \end{bmatrix}$.

**D.3. Examples from (Rantzer, 2001)**

We consider an additional experiment which includes the following system:

$$\begin{bmatrix} \dot{x}_1 \\ \dot{x}_2 \end{bmatrix} = \begin{bmatrix} -2x_1 + x_1^2 - x_2^2 \\ -2x_2 + 2x_1x_2 \end{bmatrix} \tag{32}$$

The system admits two equilibria at $(0,0)$ and $(2,0)$. As shown in Figure 7c, NLC fails to certify the stability of the system but we learn a valid density function which certify a.e. stability of the system as indicated in Figure 7d. In particular, for System (32), the region of attraction (RoA) estimated by the learned density does not extend beyond the $x \geq 2$ boundary, aligning well with the analytically known stable region.

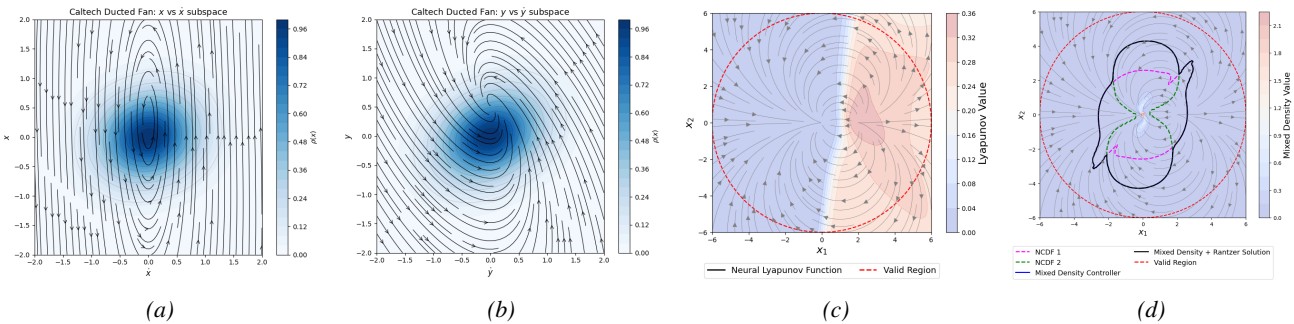

*Figure 7.* (a)-(b): shows the trajectories due to the learned controller using NCDFs in $(x, \dot{x})$ and $(y, \dot{y})$ subspace for caltech ducted fan. (c)-(d): Comparison of contours due to Neural Lyapunov Function and Neural Density Function for the system described by (32)

### D.4. Examples from (Prajna et al., 2004)

We consider the following controlled dynamical systems:

$$\dot{x}_1 = x_2 - x_1^3 + x_1^2, \qquad \dot{x}_2 = u \tag{33}$$

$$\dot{x}_1 = 2x_1^3 + x_1^2 x_2 - 6x_1 x_2^2 + 5x_2^3, \quad \dot{x}_2 = u \tag{34}$$

$$\dot{x}_1 = -6x_1 x_2^2 - x_1^2 x_2 + 2x_2^3, \qquad \dot{x}_2 = x_2 u \tag{35}$$

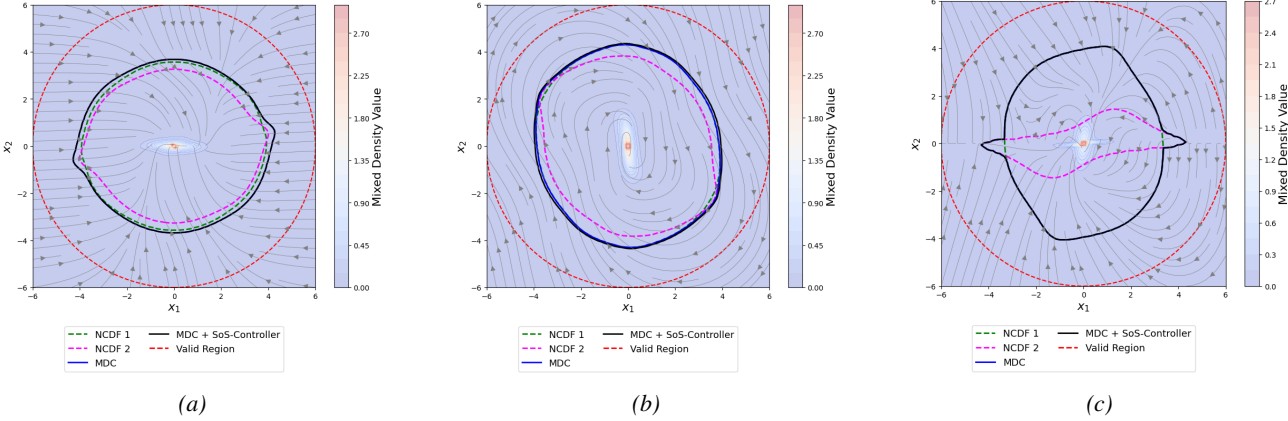

*Figure 8.* (a)-(c): Comparaison of RoAs due to the learned individual controller using NCDFs, mixed controller (MDC) and MDC blended with controller from SoS method for the system described by Equations (33) to (35)

We learn NCDFs for the system described in Equations (33) to (35). As suggested by Figure 3a, 3b and 3c, we learn a valid controller within $\|x\| \le 6$. The SoS-based controllers and densities $(u, \rho)$ for given systems are provided in (Prajna et al., 2004). We blend the learned NCDFs with the solutions provided in (Prajna et al., 2004) to achieve improved RoAs.

### D.5. Van der Pol Oscillator

We further evaluate our framework on the controlled Van der Pol oscillator. The system exhibits self-sustained oscillations arising from nonlinear damping and is governed by the following dynamics:

$$\dot{x}_1 = x_2, \quad \dot{x}_2 = \mu(1 - x_1^2)x_2 - x_1 + u, \tag{36}$$

where $x_1$ and $x_2$ denote the system states, $\mu > 0$ controls the nonlinearity and damping strength, and $u$ is the control input.

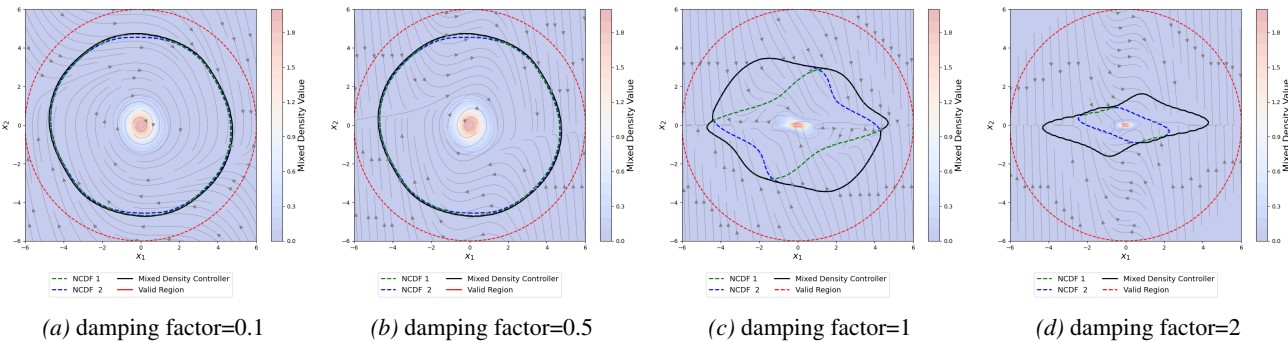

*(a)* damping factor=0.1    *(b)* damping factor=0.5    *(c)* damping factor=1    *(d)* damping factor=2

*Figure 9.* (a)–(d): Control synthesis for Van der Pol Oscillator with various damping factor

### D.6. Learning without exponential parametrization Equation (18)

To demonstrate the need for the parameterization stated in Proposition 5.1, we consider (33) for control synthesis. We parametrize the pair as follows:

$$\rho(x) = \frac{a(x)}{b(x)} \quad \psi(x) = \frac{c(x)}{a(x)} \tag{37}$$

In this setup, we assume $a(x), b(x), c(x)$ are learnable and are parameterized by neural networks. With the learner-verifier framework, we learn a non-negative density but fail to stabilize the system as indicated in the figure below:

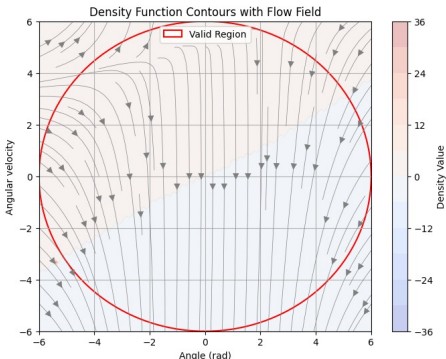

*Figure 10.* (a): Control Synthesis without exponential parametrization described by Equation (18)

### D.7. Computational Study

**Learning and Verification time:**    We provide the time to learn control-density pair for various system in the table below. We fix the falsifier call to every 100 iterations with learning rate 1e-3.

| S.N | Dynamics | Number of parameters | Time |
|---|---|---|---|
| 1 | Inverted Pendulum | A-net: 109, B-net: 96, C-net:109 | 3m 23s |
| 2 | Path Following | A-net: 13, B-net: 12, C-net:49 | 3m 11s |
| 3 | Caltech Ducted Fan | A-net: 2369,B-net: 2534, C-net:2402 | 21m 24s |

*Table 1.* Time taken to learn the control-density pair

**Total time vs Falsification Frequency:**    We use an inverted pendulum model to investigate how the total learning time for a valid control-density pair varies with the falsification frequency.

| S.N | Frequency of Falsifier | Time |
|-----|------------------------|--------|
| 1 | 100 | 3m 23s |
| 2 | 250 | 3m 53s |
| 3 | 500 | 23s |
| 4 | 750 | 22.4s |

*Table 2.* Time taken to learn the control-density pair varying the frequency the falsifier calls

The contour corresponding to inverted pendulum with various falsification frequency is provided below:

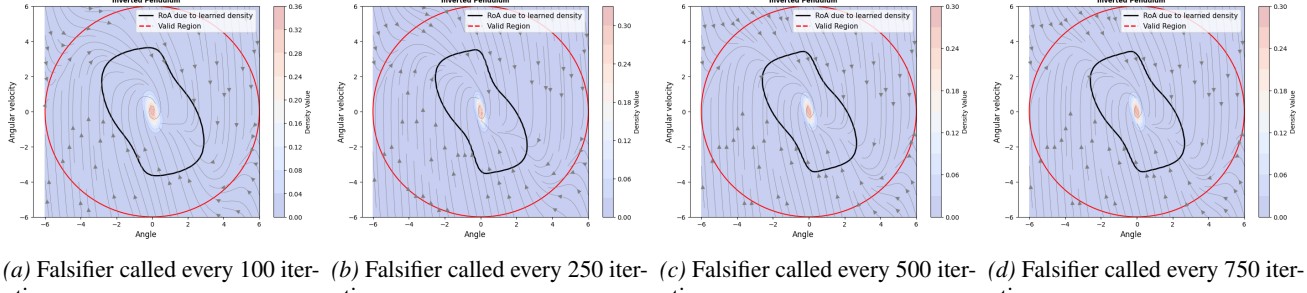

*(a)* Falsifier called every 100 iterations

*(b)* Falsifier called every 250 iterations

*(c)* Falsifier called every 500 iterations

*(d)* Falsifier called every 750 iterations

*Figure 11.* (a)–(d):Contour for Inverted Pendulum under different falsification frequency

