# OpenReview forum: "Blending Neural Control Density Functions for Stabilization and Safety"
_ICML.cc/2026/Conference — ICML 2026 regular_

### Official Review · Reviewer_dg7C · 2026-03-12

**Soundness:** 3
**Presentation:** 2
**Significance:** 2
**Originality:** 2
**Overall Recommendation:** 5
**Confidence:** 4

**Summary:**

The paper considers continuous time optimal control problems where the requirement is that the system is stabilized, has a large region of attraction, and also satisfies safety. The paper adopts a method from control theory - control density functions - as well as formal verification style of training. Theory analyzes the regions of attraction, and neural network parametrization is proposed for this kind of solution.

**Compliance With Llm Reviewing Policy:**

Affirmed.

**Final Justification:**

Updated score based on responses to my comments

**Key Questions For Authors:**

- One question I have is how many different sub-controllers N in equation (12) should be chosen. I feel that this is an important analysis point.

- I do not understand the steps from (18) to (19), can you please clarify

- Do density functions have any intuition?

- The notation of brackets in (6) and (7) seems inconsistent

**Limitations:**

- The numerical results are with small dimensional state space problems. Please clarify why that is the case, and how to deal with larger ones, where is the bottleneck?

- Figures of regions are very small and also unclear. Do red regions mean that a system is found to be unstable? Or to be more precise it cannot be guaranteed whether it is stable?

- What is the computational cost of the method? Time and memory? How does it compare with others?

**Strengths And Weaknesses:**

- The neural density function approach is not novel, borrowed from the literature, the neural network parametrization is novel.

- What is more significant is that the paper challenges the very established method of using Lyapunov functions for guaranteeing stability, and it is convincing in the numerical results. So it has the potential to replace the other methods.

- The method is combined with barrier functions, which are also widely used, so that makes it have a broader scope of problems.

- The CEGIS approach is not novel, so it is not clear whether apart from parametrization, whether this leads to novel training/verification  methods.

- One concern is that the density function does not seem to be as intuitive. The same with the divergence condition. This is a drawback of the method, where one may say I will use Lyapunov which are more intuitive, but also of the exposition of the paper, where it would be nicer to have a better demonstration of what this means.

- While the blending of controllers looks useful, I found that the theoretical analysis of ROA was not as insightful. What is the new result that is derived here and why is this complicated.

---

> ### Author Rebuttal · Authors · 2026-03-31
>
> We thank the reviewer for the constructive feedback and positive assessment, particularly the capacity for our Neural Control Density Functions to challenge the use of Lyapunov functions, and the broadness of the scope of our work. We address the concerns raised by the reviewer, such as concerns about the intuition behind density functions and clear derivation of Equation (18) from (19).
>
> **The neural density function approach is not novel, borrowed from the literature, the neural network parametrization is novel**
>
> While density functions have been studied before, ours is the first work to study learning density functions using neural networks with the proposed novel parameterization (given in Proposition 4.1).
>
> **The CEGIS approach is not novel, so it is not clear whether apart from parametrization, whether this leads to novel training/verification methods.**
>
> We agree that the CEGIS approach is well studied in the literature. However, this work is the first to apply it to the problem of learning control density functions for stabilization, safety, and to smoothly blend controllers.
>
> **One concern is that the density function does not seem to be as intuitive… it would be nicer to have a better demonstration of what this means.**
>
> While some users may find density functions to be less intuitive than using Lyapunov functions, using density functions provides additional benefits, including, as the reviewer noted, the ability to certify almost-sure stability in the presence of multiple equilibria, and the ability to use density functions to smoothly blend controllers.
>
> ** While the blending of controllers looks useful… What is the new result that is derived here and why is this complicated.**
>
> Theorems 3.1 and 3.2 are the first works to analyze the regions of attraction of smoothly blended controllers by way of density functions, and are the first to show that the region of attraction of a blended controller contains the individual ROAs of the constituent controllers. Thus, they give a formal guarantee of improvement of the ROA achieved by smoothly blending controllers with density functions.
>
> **One question I have is how many different sub-controllers N in equation (12) should be chosen. I feel that this is an important analysis point.**
>
> The number of sub-controllers is a free design parameter and can be chosen by the practitioner. Most importantly, as we show in Theorems 3.1 and 3.2, blending any number of controllers always leads to an improvement in the region of attraction.  In particular, adding additional sub-controllers does not degrade the guarantee and can only expand the overall certified region when appropriate local controllers are provided.
>
> **I do not understand the steps from (18) to (19), can you please clarify**
>
> Equation (19) is obtained by substituting the parameterization introduced in (18) into Eq. (9) and simplifying the resulting expression. For clarity, we have added a detailed step-by-step derivation for the reviewer’s convenience, provided in the section “Explanation of (18) to (19)” at [ https://anonymous.4open.science/r/Neural-Control-Density-Functions-5F32/ICML_Rebuttal/additional_doc.pdf ].
>
> **Do density functions have any intuition?**
>
> Density functions can be thought to describe how mass flows through the state space. The condition that density function is positive and integrable implies the mass is generated everywhere and the divergence condition ensures the mass must flow into a sink [1].
>
> **The notation of brackets in (6) and (7) seems inconsistent**
>
> We thank our reviewer for pointing out this typo and have corrected the modified manuscript.
>
> ## References
>
> [1] Meinsma, 2006. On Rantzer’s density function

---

> > ### Author Rebuttal · Reviewer_dg7C · 2026-04-02
> >
> > Thank you for your response. Would you please clarify my earlier comments:
> >
> > - Size of state space
> > - Computational cost
> > - How to read the figures

---

> > > ### Author Response · Authors · 2026-04-02
> > >
> > > We thank the reviewer for the follow-up questions, and apologize for missing their concerns before. We address their concerns now below. We provide a computational cost table for our method, additional experiments on higher-dimensional systems, and a clarification on how to interpret the figures. Kindly let us know if we have satisfactorily clarified your concerns.
> > >
> > > **Size of state space**
> > >
> > > We thank the reviewer for the insightful question about scalability and provide experiments demonstrating that our neural control barrier function approach is highly effective on higher-dimensional systems to address their concern. Specifically, we evaluate the Neural Control Density Function approach on the Caltech Ducted Fan and PVTOL dynamics — both 6-dimensional systems with 2-dimensional control inputs that are standard higher-dimensional systems used in recent publications in this space, such as [2,3] — demonstrating that Neural Control Density Functions can be effectively applied to these systems.  Detailed results are provided in the section titled ‘’Additional higher dimension experiments’’ at [ https://anonymous.4open.science/r/Neural-Control-Density-Functions-5F32/ICML_Rebuttal/additional_doc.pdf ].
> > >
> > > **Computational Cost**
> > >
> > > In practice, we find that the training time is comparable to Neural Lyapunov Control [1], and the computational bottleneck is the SMT falsifier- a bottleneck shared by both methods. For example, learning a Neural Control Density Function for the Inverted Pendulum system takes around 25s as reported in [1], whereas our method requires 22.4s. We include the computational study [where we study time taken to learn Neural Control Density Functions] in the section titled “Computational Study’’ available in the link above, and will be incorporated into the appendix of the revised draft. The peak memory usage during training for our approach for the Inverted Pendulum control problem is 17.48 MB, whereas it is 17.11 for Neural Lyapunov Control [1].
> > >
> > >
> > >
> > > **How to read the figures**
> > >
> > > We would like to clarify the figures presented in Section 5. The **red dashed circle** in each figure denotes the *valid region* $S=\\{ x:||x||\leq r \\} $ [for some $r$, usually 6 in our experiments] over which the data was provided and the density certificate was formally verified by the SMT solver. It does not indicate instability. Points outside the red circle are simply outside the verified domain — no claim of stability or instability is made there. Similarly, the lines mentioned in the legend indicate the inner approximation of RoA due to the method mentioned in the legend (such as ‘Lyapunov Controller’ or ‘Mixed Density’ in Figure 3a). The different colours of the contours in the plot indicate the values taken by the density function where the maximum is attained at the point of origin [or point of stability in the experiments considered].
> > >
> > > The black regions in Figure 4 indicate the unsafe set, and as shown in the figure, the trajectories generated with the learned controller avoid the highlighted unsafe set.
> > >
> > > ### References
> > >
> > > [1] Chang et al, 2019. Neural Lyapunov Control.
> > >
> > > [2] Li et al, 2025. Two-Stage Learning of Stabilizing Neural Controllers via Zubov Sampling and Iterative Domain Expansion
> > >
> > > [3] Wang et al, 2024. Lyapunov-stable Neural Control for State and Output Feedback: A Novel Formulation

---

### Official Review · Reviewer_dNtn · 2026-03-12

**Soundness:** 2
**Presentation:** 3
**Significance:** 2
**Originality:** 2
**Overall Recommendation:** 4
**Confidence:** 3

**Summary:**

This paper proposes a neural control framework based on density functions as stability certificates. Instead of relying on Lyapunov functions, the authors employ the density theorem introduced by Rantzer to certify almost-everywhere asymptotic stability. The main contribution of the paper is a framework that enables blending multiple neural controllers while preserving stability guarantees. The authors further extend the method to incorporate safety constraints and demonstrate the approach on several nonlinear control examples.

**Compliance With Llm Reviewing Policy:**

Affirmed.

**Final Justification:**

The rebuttal addressed my major issues, so I raise the score accordingly.

**Key Questions For Authors:**

Q1: Since Lyapunov methods provide stronger stability guarantees but weaker compositional properties, while density certificates provide the opposite trade-off, could the authors comment on whether hybrid approaches combining both certificates might be possible?

Q2: In the first experiment in Figure 1, could the authors clarify why a controller is learned rather than directly certifying the stability of the original system?

**Limitations:**

Yes

**Strengths And Weaknesses:**

**Strengths**

1. The paper leverages the density stability theorem as an alternative to Lyapunov-based methods. This is an underexplored and interesting direction in neural control and provides a fresh perspective on stability certification.

2. The most interesting aspect of the paper is the controller blending result. The authors show that controllers equipped with density certificates can be blended while preserving stability guarantees, and that the resulting region of attraction contains the union of the original regions. This property is difficult to obtain using classical Lyapunov-based neural control methods.

3. The proposed density based framework can certify stability in systems with multiple equilibria and saddle nodes.

**Weaknesses**

1. The density-based approach guarantees almost-everywhere asymptotic stability, rather than stronger notions such as exponential stability. So its hard to estimate the convergence time in practice.

2. The experimental section demonstrates feasibility but remains relatively simple. Additional experiments on higher-dimensional or more complex systems would strengthen the empirical validation.

---

> ### Author Rebuttal · Authors · 2026-03-31
>
> We thank the reviewer for the thoughtful feedback and for highlighting the novelty of using density functions and the controller blending result. We address the concerns and questions below, and provide additional experiments on higher-dimensional systems as requested by the reviewer.
>
> **The density-based approach guarantees almost-everywhere asymptotic stability… its hard to estimate the convergence time in practice.**
>
>  While estimating convergence time is indeed of practical interest, we note that several recent works in this area (e.g., [2],[3]) also do not provide guarantees on exponential stability or explicit convergence rates. Instead, these methods certify asymptotic stability, which similarly does not yield tractable bounds on convergence time.
> Our density-based approach therefore aligns with the guarantees provided by commonly used baselines, while offering additional flexibility in handling systems with multiple equilibria and enabling blending of the controllers.
>
> **The experimental section demonstrates feasibility but remains relatively simple. Additional experiments on higher-dimensional or more complex systems would strengthen the empirical validation.**
>
> We disagree that our experiments are relatively simple. We demonstrate the effectiveness of neural density control across a range of settings, including systems with multiple equilibria, standard nonlinear benchmarks such as the inverted pendulum, path-following tasks, and safe control synthesis in environments with multiple obstacles. That said, we agree that experiments on higher-dimensional systems would further strengthen empirical validation. To address this, we include additional experiments on the Caltech ducted fan and PVTOL dynamics, with results provided in section titled ‘’Additional higher dimension experiment’’ at [https://anonymous.4open.science/r/Neural-Control-Density-Functions-5F32/ICML_Rebuttal/additional_doc.pdf].
>
> **Since Lyapunov methods provide stronger stability guarantees but weaker compositional properties… could the authors comment on whether hybrid approaches combining both certificates might be possible?**
>
> As noted in Proposition 1 in [1], for appropriate $\alpha>0$ we can consider $\rho(x)=V(x)^{-\alpha}$ which allows a Lyapunov-based controller certificate to be blended directly with a density-based controller certificate. We note that this connection is discussed in the experiments section under the paragraph titled *Blending Neural Lyapunov Controllers*, where we demonstrate empirically that such blending yields larger certified regions of attraction than either certificate achieves individually. Moreover, Theorem 3.2  provides the first rigorous guarantee of the domination of a blended controller in this setting.
>
>
>
> **In the first experiment in Figure 1, could the authors clarify why a controller is learned rather than directly certifying the stability of the original system?**
>
> In figure 1, no controller is learned - the experiment is purely a stability certification task, wherein we learn only a density function to certify the almost everywhere stability of a system with multiple equilibria, and show that Lyapunov functions fail in this setting.
>
> ### References
>
> [1] Rantzer 2001. A Dual to Lyapunov’s Stability Theorem.
>
> [2] Chang et al, 2019. Neural Lyapunov Control.
>
> [3] Li et al, 2025. Two-Stage Learning of Stabilizing Neural Controllers via Zubov Sampling and Iterative Domain Expansion

---

### Official Review · Reviewer_yQ75 · 2026-03-13

**Soundness:** 3
**Presentation:** 2
**Significance:** 2
**Originality:** 2
**Overall Recommendation:** 4
**Confidence:** 2

**Summary:**

This paper proposes a novel Neural Control Density Functions (NCDFs) method for stability analysis and safety control synthesis of nonlinear systems. Compared to existing neural Lyapunov control (NLC) methods, density functions can prove stability almost everywhere, making them effective in systems with unstable equilibrium or saddle points. In addition, this article theoretically proves that after using density function to smoothly fuse multiple controllers, the closed-loop system's Regoin of Attractions must include the union of all single controller's Regoin of Attractions. The experiment shows that the hybrid controller outperforms NLC and existing advanced methods based on Sum of Squares in the attraction domain.

**Compliance With Llm Reviewing Policy:**

Affirmed.

**Final Justification:**

The authors addressed my questions in the final response, so I raise my score accordingly.

**Key Questions For Authors:**

See the weakness above.

**Limitations:**

yes

**Strengths And Weaknesses:**

### Soundness

The theoretical derivation of the paper is rigorous, the method design is rigorous, and it has been theoretically validated. The experiment fully demonstrated the practical effect of the fusion controller in expanding the stability domain and verified the correctness of the theoretical derivation.

Weakness: Lack of quantitative evaluation of computational costs. The author also mentioned in the limitations section that the proposed parameterization method requires joint optimization of three neural networks, but did not present results related to training time and counter example verification time. Additionally, the verification frequency setting lacks ablation analysis. Due to the current method using a lower validation frequency compared to the previous NLC method, it is necessary to explore the impact of using different validation frequencies on training stability and performance. Or is it due to the high cost of computational time?

### Presentation

The drawing of the figure is not satisfactory. The size and title format are not consistent, the horizontal and vertical axis label formats of fig.2 are not consistent, and there are unknown links in the chart caption. In addition, the font size in the chart is too small, which affects reading. The author should re-examine the standardized drawing of all figures.

### Significance

The Lyapunov method cannot prove the global or large-scale stability of systems with multiple equilibrium points, and different controllers cannot be smoothly fused directly. The method presented in this article provides a feasible and formally guaranteed modern deep learning solution for the classic pain points mentioned above.

Weakness: Finding a globally effective density function in high-dimensional complex systems can be very difficult

### Originality

The contribution of this article lies in the organic combination of neural networks, CEGIS framework, and density functions, which expands the research boundary of Neural Control Certification.

---

> ### Author Rebuttal · Authors · 2026-03-31
>
> We thank the reviewer for the careful reading, constructive feedback, and for highlighting the strengths of our work, particularly the rigor of the theoretical derivations and the effectiveness of the blended controller in expanding the certified region of attraction. We address the reviewers' concerns about density functions, as well as ablation studies and additional experiments on high dimensional systems.
>
> **Lack of quantitative evaluation of computational costs... Additionally, the verification frequency setting lacks ablation analysis.**
>
>
> We note that Neural Lyapunov Control methods also require joint optimization of two networks (a controller and a Lyapunov function), so our method introduces only one additional network relative to the baseline. In practice, we find that the training time is comparable to NLC and the computational bottleneck is the SMT falsifier- a bottleneck shared by both methods. For example: Inverted Pendulum takes around 25s as reported in [1] and our method requires 22.4s. We provide a table comparing training time and falsification frequency in section titled ‘’Computational Study‘’ at [  https://anonymous.4open.science/r/Neural-Control-Density-Functions-5F32/ICML_Rebuttal/additional_doc.pdf ]. In addition, we include a computational study summarizing the time required to learn density-function pairs across several nonlinear control problems. We will add this ablation and timing analysis to the appendix of the revised draft.
>
> **Due to the current method using a lower validation frequenc… Or is it due to the high cost of computational time?**
>
> In the CEGIS approach, verification (using dReal, or $\alpha,\beta$-CROWN) is a crucial bottleneck in terms of time consumption. In our work, we show that we can achieve high performance with minimal verifier call frequency.
> To address the reviewer’s concern, we ran an ablation study varying the falsification frequency in inverted pendulum [Eq-31 in the paper] and report the training time and contour plots in ‘’Computational Study‘’ at [  https://anonymous.4open.science/r/Neural-Control-Density-Functions-5F32/ICML_Rebuttal/additional_doc.pdf ]. The results confirm that solution quality is largely insensitive to this choice across the range considered, while training time decreases at lower frequencies.  We will add this ablation to the appendix of the revised draft.
>
> **Weakness: Finding a globally effective density function in high-dimensional complex systems can be very difficult**
>
> We point out that in this work, and in all CEGIS based approaches for control with neural networks, [1,2,3,4,5], we principally aim to find controllers that are verifiably stabilizing on a compact set. In this regard, owing to our novel exponential parameterization (Proposition 4.1),  learning neural control density functions is no more difficult than finding neural control Lyapunov functions.
> However, in the general setting, synthesizing stabilizing controllers using density functions is often more tractable than doing so with control lyapunov functions (CLFs), as the set of CLF-controller pairs is generally non-convex and non-connected (see [7]). As such, they cannot be optimized over, whereas using density functions avoids this challenge (as noted in [6])
>
> ### References
>
> [1] Chang et al, 2019.Neural Lyapunov Control
>
> [2] Yang et al, 2024.  Lyapunov-stable neural control for state and output feedback: A novel formulation.
>
> [3] Liu et al, 2023. Safe control under input limits with neural control barrier functions.
>
> [4] Dawson et al, 2023. Safe control with learned certificates: A survey of neural lyapunov, barrier, and contraction methods for robotics and control.
>
> [5] Zhou et al, 2022.  Neural lyapunov control of unknown nonlinear systems with stability guarantees.
>
> [6] Rantzer, 2001. A Dual to Lyapunov’s Stability Theorem.
>
> [7] Prieur and Praly, 1999. Uniting Local and Global Controllers.

---

> > ### Author Rebuttal · Reviewer_yQ75 · 2026-04-04
> >
> > Thank you for the clarification. The response is helpful, and I strongly recommend the authors to include these results in the final manuscript. But I still need to notice that the author should also pay more attention to their presentation in the manuscript. I prefer to maintain my score.

---

> > > ### Author Response · Authors · 2026-04-07
> > >
> > > We thank the reviewer for their acknowledgment, and are happy that we have addressed their key concerns. We will now discuss modifications to our presentation in Section 6 below, based on the reviewer's feedback.
> > >
> > > ---
> > >
> > > ### **The size and title format are not consistent, the horizontal and vertical axis label formats of fig.2 are not consistent, and there are unknown links in the chart caption.**
> > >
> > > We have modified the figures to address this concern, and corrected the links in the captions as well. We also clarify that Figure 2a is for NCDFs learned for stabilizing the system in Eq. 25, whereas Figure 2b is for the system in Eq. 26. We provide snapshots of the amended Figure 2 in:
> > >
> > > [ https://anonymous.4open.science/r/Neural-Control-Density-Functions-5F32/Updated_Plots/Screenshot_7-4-2026_212043_.jpeg ]
> > >
> > > ---
> > >
> > > ### **In addition, the font size in the chart is too small, which affects reading.**
> > >
> > > We have modified the figures in our manuscript to address their concerns. Specifically, we increased and standardized the font sizes in axis labels and legends, and moved the legend below the figure. We provide snapshots of the new figures below.
> > >
> > > **Modified Figure 1:**
> > >  [ https://anonymous.4open.science/r/Neural-Control-Density-Functions-5F32/Updated_Plots/Screenshot_7-4-2026_212027_.jpeg ],
> > >
> > > **Modified Figure 2:**
> > > [ https://anonymous.4open.science/r/Neural-Control-Density-Functions-5F32/Updated_Plots/Screenshot_7-4-2026_212043_.jpeg ],
> > >
> > > **Modified Figure 3:**
> > > [ https://anonymous.4open.science/r/Neural-Control-Density-Functions-5F32/Updated_Plots/Screenshot_7-4-2026_212055_.jpeg ],
> > >
> > > **Modified Figure 4:**
> > > [ https://anonymous.4open.science/r/Neural-Control-Density-Functions-5F32/Updated_Plots/Screenshot_7-4-2026_212116_.jpeg ],
> > >
> > > **Equations Corresponding to Figure 2 (links fixed):**
> > > [
> > > https://anonymous.4open.science/r/Neural-Control-Density-Functions-5F32/Updated_Plots/Screenshot_7-4-2026_212143_.jpeg  ].
> > >
> > > ---
> > >
> > >
> > > We hope that we have adequately addressed the reviewers' concerns so that they increase their score, especially since we have addressed the other questions from their review, as noted in their acknowledgment. If there are any other concerns diminishing the reviewer’s appraisal of our work, we are eager to engage with them further.

---

### Official Review · Reviewer_hvgG · 2026-03-17

**Soundness:** 2
**Presentation:** 3
**Significance:** 2
**Originality:** 3
**Overall Recommendation:** 3
**Confidence:** 4

**Summary:**

This paper proposes the use of control density functions as a framework for constructing stability certificates and synthesizing neural networks controllers which are asymptotically stable almost everywhere, relaxing the usual point-wise conditions of neural Lyapunov certificates. This paper extends the results of control density functions to blend several a.e. Stable controllers to improve the volume of the effective region of attraction under certain conditions. The paper experimentally verifies the framework by parameterizing and learning density functions that can be blended together to result in increased regions of certified stability.

**Compliance With Llm Reviewing Policy:**

Affirmed.

**Final Justification:**

I believe the idea of verifying almost-everywhere regions of attraction of neural closed-loops under the Rantzer density argument is quite novel and that the way the learned controller is parameterized and can be blended is very interesting.

However, I'm still not satisfied with the author's explanation about how the divergence of the blended closed-loop controller is *certified* via falsification. In the rebuttal, the authors point to a proposed loss function which should encourage the divergence term of a single controlled closed-loop to be non-positive. Not only is this regularization not sufficient for certification, but it doesn't explain why the blended closed-loop $F^\*$ will have negative divergence. I think this is very important to clarify in supporting one of the main claims of paper (controller blending), especially when comparing against works that do formally certify the region of attraction.

I would strongly encourage the authors to make this clear and reattempt submitting the paper.

**Key Questions For Authors:**

- The blended controller theorem of 3.1 seems to require that the $D_i$ covers $Q_i$ except for measure zero, but I don't see how that is practically verified in the experiments at any point, yet the blending argument is applied. Can you clarify whether or not these are hard certificates or more like heuristic arguments? If you arent formally satisfying the assumptions of theorem 3.1, I don't see any merit to blending the controllers and comparing to SoS and density functions without stronger empirical evidence. Clarification would be greatly appreciated.

**Limitations:**

yes

**Strengths And Weaknesses:**

Pros

- Control density functions for learning-based stability analysis and controller synthesis has not been explored before and is quite interesting. The idea of certificates holding almost everywhere seems quite promising for practical systems and could relax the usual point-wise certificates Lyapunov.
- The result claiming blended certified densities can result in a larger almost everywhere ROA is an interesting and potentially useful control theoretic result.

Cons

- Lack of experiment comparisons: In figure 2a it is claimed that “the neural Lyapunov method failed to verify”, which I assume means the method by Chang et al 2019. However there are several other methods cited (Yang 2024) that do provide large certified ROA for systems greater than 2D (and not cited, [Li2025], [Wang2024]). Even though these are not density/measure arguments, their certification is actually a stronger result, so a density method should enhance the ROA volume greatly compared to their pointwise arguments. This would make the paper much more convincing.

- It seems that the Rantzer density argument is a global one over all of $R^n$, however your verification analysis must be done over a bounded set S. So as in Theorem 5.1, the extra constraint forward invariance constraint on C is required. However, when you perform your verification in practice, you must guarantee that C is contained in the region of interest S that you verify, or else the trajectory might leave the region you checked eq (23) over. I don’t see how you guarantee the levelset of the barrier function $h$ are contained in S.

- There is something about verification of the blended controller experimental results that appears heuristic (see question below). The paper is not very clear in its positioning, whether these results are hard guarantees like Yang 2024 or simply approximations.

Minor
- Definition 2.1 is asymptotic stability, not Lyapunov stability. Lyapunov stability does not imply convergence of trajectories to an equilibrium.
- Definition 2.2, The condition $F_t(x_0)$ in $D$ for all $x_0$ in $D$ is redundant (perhaps you're just mentioning this property for later?). The largest set D s.t. $\lim_t F_t(x_0) = x^\*$ for all $x_0$ implies forward invariance of $D$. If at some point $F_t(x_0)$ leaves $D$, then that point should also belong to $D$ or else it won’t converge to $x^\*$.

References

- [Li2025] Two-Stage Learning of Stabilizing Neural Controllers via Zubov Sampling and Iterative Domain Expansion, neurips 2024

- [Wang2024] Actor-Critic Physics-Informed Neural Lyapunov Control, IEEE

---

> ### Author Rebuttal · Authors · 2026-03-31
>
> We thank the reviewer for the thoughtful and detailed feedback, and for highlighting the strengths of our work, particularly the novelty of control density functions for learning-based stability analysis, and the theoretical result on blending controllers to enlarge the region of attraction (ROA). We address the concerns below.
>
> **Lack of experiment comparisons**
>
> We believe that our empirical comparisons are very thorough, as we evaluate our method on a variety of tasks including controller blending, safety, and stability analysis with multiple equilibria. We also believe the reviewer is referring to Figure 1 rather than Figure 2, as it is in Figure 1 that we discuss the failure of Neural Lyapunov methods to certify stability in the presence of multiple equilibria [1], which can be addressed with density functions. Hence, our experiment studied stability analysis for systems containing both stable and unstable equilibria . [2], [3], [4],[5] rely on Lyapunov-based certificates, which face difficulties in such settings and thus can not be used to certify the stability of the system, let alone certify a large RoA.
>
> Our intent in Figure 1 is therefore not to provide an exhaustive comparison with all recent Lyapunov based methods, but to highlight this limitation and demonstrate that Neural control density function can certify a.e. stability in cases where standard Neural Lyapunov approaches fail.
>
> **Even though these are not density/measure arguments, their certification is actually a stronger result…  This would make the paper much more convincing.**
>
> We point out that the ROAs certified in the cited papers would indeed always be subsumed by the density function ROAs. Moreover, the methods in [3] and [4] are not directly applicable to some of the systems we consider, as these systems contain multiple equilibria. Such settings pose inherent challenges for Lyapunov-based certificates, and these approaches also do not naturally support smooth blending of multiple controllers, which is a central component of our framework. We also empirically demonstrated that the RoAs obtained using the mixed density controller is larger than that of Neural Lyapunov Control, further supporting the practical benefits of our approach.
>
>
>
> **It seems that the Rantzer density argument is a global one … I don’t see how you guarantee the levelset of the barrier function h are contained in S.**
>
> We thank the reviewer for this insightful comment. While the original Rantzer density argument is global, extensions exist that provide a.e. stability guarantees over bounded sets. In particular, [6, Theorem 1] establishes forward invariance and a.e. stability for a bounded region (S). We will revise the paper to explicitly reference this result and clarify how it applies in our setting (we have also provided a brief write-up in the section Safe Stabilization with Neural Control Density Functions in [  https://anonymous.4open.science/r/Neural-Control-Density-Functions-5F32/ICML_Rebuttal/additional_doc.pdf ]).
> Because this formulation is local to the bounded set (S), it allows us to ensure that the relevant level sets of the barrier function (h) are contained within (S). We will therefore amend Theorem 5.1 to incorporate this local stability argument following [6]. In our experiments, we ensure by construction that the safe set (C) lies within the region of interest (S), so that trajectories remain in the domain where the verification conditions (Eq. (23)) are enforced.
>
> **Minor issues:**
>
> We thank the reviewer for the careful reading, and have made the following amendments:
>
> 1. We have amended the definition to state asymptotic stability
>
> 2. We have amended Definition 2.2 to include the condition $F_t(x_0) \in D$ for all $x_0\in D$ and $t > 0$.
>
> **The blended controller theorem of 3.1 seems to require that ...  Clarification would be greatly appreciated.**
>
> We point out that Theorem 3.2 formally guarantees that the certified region of attraction for the blended controller contains all the regions of attraction of the constituent controllers in normal cases, hence establishing the merit of blending controllers. Our experiments also showcase the improvement in ROA achieved by blending controllers, and that they always outperform single controllers.
>
> ## References
>
> [1] Rantzer, 2001. A Dual to Lyapunov’s Stability Theorem
>
> [2] Chang et al, 2019. Neural Lyapunov Control
>
> [3] Li et al, 2025. Two-Stage Learning of Stabilizing Neural Controllers via Zubov Sampling and Iterative Domain Expansion
>
> [4] Wang et al, 2024. Actor-Critic Physics-Informed Neural Lyapunov Control
>
> [5] Zhou et al, 2022.  Neural lyapunov control of unknown nonlinear systems with stability guarantees.
>
> [6] Masubuchi, 2006. Stability Analysis and Stabilization of Nonlinear Systems via Locally Defined Density Functions

---

> > ### Author Rebuttal · Reviewer_hvgG · 2026-04-04
> >
> > Thank you for the clarification and addressing most of my concerns. I'm glad to see that the local version of the Rantzer density argument clarified. Indeed the experimental cases of systems with multiple equilibria are quite interesting.
> >
> > I see now that although Theorem 3.1 requires that  $ D_i = \\{ x \in S : \rho_i(x) \geq \tau_i \\}$ approximates the true ROA up to measure zero, Theorem 3.2 only requires $D_i \subset Q_i$. However, Theorem 3.2 involves  the additional condition that the closed-loop dynamics have a negative divergence $\nabla \cdot F^\* < 0$. **This does not seem to appear in the Falsification condition of eq 20 or 23**. Can you explain clearly how this is introduced in the certification?
> >
> > Additionally, $\nabla \cdot F^\* < 0$ is a fairly strong condition for a closed-loop control system. Can you explain how realistic this assumption is and what are the limitations? Are there obstructions for this assumption in, for example, a classical swing-up pendulum problem with torque constraints (ie you have to pump energy in before swinging up) or for under-actuated systems common in robotics?
> >
> >
> > Regarding the minor comment about the ROA definition, you replied
> >
> > > We have amended Definition 2.2 to include the condition $F_t(x_0) \in D$ for all $x_0\in D$ and $t > 0$
> >
> > You actually already have that condition in the definition (minus the quantifier $t >0$). What I suggested in my comment was that you don't need the above as a condition, but that it already follows by the definition of "the largest set $D$ containing $x^\*$ such that $\lim_{t\rightarrow \infty} F_t(x_0) = x^\*$ ".
> >
> >
> > Lastly, it is pointed out that there are several difficulties of Lyapunov-based certification due to notion of stability being more strict compared to the density results presented here. However, I think it is interesting to see how this density-based certified training method compares to Lyapunov-based even in the established settings. **I still think that that the results for Inverted pendulum and path tracking in Figure 2 do not represent the state-of-the-art given by Lyapunov-based learning certificates found in Yang2024 (see figure 5)**, which show significantly larger ROA than the Lyapunov baselines provided.
> >
> >
> > [Yang2024] Yang et al, Lyapunov-stable Neural Control for State and Output Feedback: A Novel Formulation, ICML 2024

---

> > > ### Author Response · Authors · 2026-04-07
> > >
> > > We thank the reviewer for their insightful follow-up questions. We address them below.
> > >
> > > ### Where the divergence sign is certified
> > >  We have discussed how the sign of the divergence is ensured in the main body of our paper, on lines 230-236 in Section 4.2. Specifically, when training neural control density functions, the sign condition is ensured by adding a penalty for the sign of the divergence of the closed-loop vector field as follows:
> > >
> > > $L_{\zeta}'(\theta)=L_\zeta(\theta)+\mathbb{E}_{x \sim \mathcal{\zeta}(S)}[\max(0,\nabla\cdot (f(x)+g(x)u(x)))]  $
> > >
> > > ---
> > >
> > > ### Strength of the Assumption of the Divergence Sign
> > > First, we note that Theorem 3.2 requires the condition $\nabla \cdot F(x) < 0$ in order to be able to construct Lyapunov functions from the learned neural density function, and allows for stronger guarantees on the ROA when blending controllers. In practice, this is achieved by enforcing the sign of the closed loop system’s divergence as we mention above.
> > >
> > > Second, we note that while the negative divergence assumption is a moderately strong one, it does appear in a variety of real-world and robotics systems, or can be easily achieved by design. We provide examples below
> > >
> > > **Swing-up Pendulum with Friction and Torque Constraints**  We consider dynamics with $x= (\theta,\omega)$ and $\dot{\theta} = \omega$, and $\dot{\omega} = \tau -  b\omega/ml^2 - (g/l)\sin(\theta)$, with control input $\tau$ which satisfies $|\tau| < \tau^\star$ [1]. The divergence can be easily computed to be $\nabla \cdot F(x) = -b/(ml^2) + \partial \tau/\partial \omega$. In the control density function setting, we add the constraints $-\tau^\star\rho(x) \leq \psi(x) \leq \tau^\star\rho(x)$ (by substituting  $u = \tau = \psi(x)/\rho(x)$) and $b/(ml^2) \geq (1/\rho(x)) (\frac{\partial}{\partial \omega}(\psi(x) - \tau^\star\rho(x))$ (by substituting the two-sided constraint into the divergence of the closed-loop system). If a controller satisfying these constraints is synthesized using NCDFs, then it are guaranteed that the closed-loop divergence is negative.
> > >
> > > **Gradient Flow methods for Navigation:** We consider Navigation function based methods [2] , where the closed loop dynamics are $\dot{x} =F(x) =  u= -K\nabla \phi(x)$, where $\phi$ is a potential function that is minimized at the destination. Convex potentials are commonly used [3], and these guarantee that the $\nabla \cdot F(x) < 0$.
> > >
> > > **Inverse Dynamics Control of Robotic Manipulators** Consider the system $M(q)\ddot{q} + b(q,\dot{q}) +g(q) = \tau$, where $q$ is the position vector and the torque $\tau$ is the control input [2]. We can write the closed loop dynamics given a PD controller, with  $x = (q,\dot{q})$, as $\dot{x} = F(x) = [\dot{q}, K_P(q-q^\star) - K_D\dot{q}]$. By inspection, we have $\nabla \cdot F(x) = -\mathrm{Tr}(K_D) < 0$, since $K_D$ is necessarily PSD.
> > >
> > > ---
> > >
> > > ### Experimental comparison with Yang et al, 2024
> > >
> > > We conducted experiments comparing our work with that of Yang et al, 2024. Plots of our experiments are provided at: [https://anonymous.4open.science/r/Neural-Control-Density-Functions-5F32/rebuttal/Plots_yang.pdf ] and the notebook corresponding to these experiments are available in the same folder. We observe that our approach yields a far larger ROA on the set $\\{x:\|x\|^2 \leq 36\\}$ in the inverted pendulum case. On the path following problem, on the set $\\{x: -3\leq x_i\leq 3\\}$, our approach yields ROAs that are at least comparable to that obtained by the method in [Yang2024].
> > >
> > >  —
> > >
> > > If the reviewer has any additional concerns or questions about our work, we are eager to engage with them further.
> > >
> > > —
> > >
> > > ### References
> > >
> > > [1] Russ Tedrake. Underactuated Robotics (lecture notes)
> > >
> > > [2] M. W. Spong et al.  Robot Modeling and Control
> > >
> > > [3] Paternain et al. Navigation Functions for Convex Potentials in a Space with Convex Obstacles
> > >
> > > [Yang2024] Yang et al, Lyapunov-stable Neural Control for State and Output Feedback: A Novel Formulation, ICML 2024

---

### Decision · Program_Chairs · 2026-04-30

**Decision:**

Accept (regular)

**Comment:**

Reviewers broadly recognized the novelty and value of the exponential parameterization of density functions that provably satisfies the integrability condition, the theoretical result showing that blended controllers achieve regions of attraction containing the union of the constituent regions (see Theorem 3.1), and the demonstration that density-function-based certification succeeds on systems with multiple equilibria where Neural Lyapunov methods fail. Two reviewers raised their scores after the rebuttal, and the paper addresses a genuine gap in the neural certificate literature.

However, one reviewer raised a serious and unresolved technical concern that must be addressed in the final version.

Theorem 3.2 requires that the blended closed-loop system $F^\star$ satisfy $\nabla \cdot F^\star < 0$, which is necessary for the stronger RoA guarantee. This condition is never formally verified by the CEGIS falsifier. The authors pointed to a loss penalty encouraging negative divergence of each individual constituent vector field $F_i$ ,but this is insufficient as the divergence of the blended field $F^\star(x) = \sum_i \tau_i(x) F_i(x)$ involves cross-terms $\nabla \tau_i(x) \cdot F_i(x)$ through the product rule, which have no guaranteed sign even when each $\nabla \cdot F_i < 0$ is enforced. The authors' response did not address this gap, and the reviewer explicitly restated the concern after reading the rebuttal, remaining unsatisfied.

This gap propagates to the empirical evaluations, leading to unfair comparisons between formally verified baselines and only informally verified certificates in the paper. Specifically, the claim that blended controllers achieve formally certified RoAs superior to Yang et al. 2024 is undermined by the unverified $\nabla \cdot F^* < 0$ condition. The rebuttal plots further evaluate Yang et al.'s method on a domain of $|x| \leq 6$ rather than the larger domain in Yang et al.'s own paper, where their certified RoA fills nearly the entire verified region, making it difficult to ascertain if the comparison is indeed fair or not.

We recommend acceptance on the strength of the paper's theoretical contributions and the novelty of the density function parameterization, but with the strong requirement that the final version either extend the CEGIS falsification to formally verify $\nabla \cdot F^* < 0$ for the blended system, or clearly state that Theorem 3.2's premise is enforced only approximately via regularization and not certified. The comparison with Yang et al. should be conducted on the same domain used in that paper, with explicit acknowledgment of the distinction between formally certified and heuristically enforced RoA bounds.